# LOVASZ THETA CONTRASTIVE LEARNING

## ABSTRACT

We establish a connection between the Lovasz theta function of a graph and the widely used InfoNCE loss. We show that under certain conditions, the minima of the InfoNCE loss are related to minimizing the Lovasz theta function on the empty similarity graph between the samples. Building on this connection, we generalize contrastive learning on weighted similarity graphs between samples. Our Lovasz theta contrastive loss uses a weighted graph that can be learned to take into account similarities between our data. We evaluate our method on image classification tasks, demonstrating an improvement of $1\%$ in the supervised case and up to $4\%$ in the unsupervised case.

## 1 INTRODUCTION

The Lovasz theta function is a fundamental quantity in graph theory. It can be considered as the natural semidefinite relaxation of the graph independence number and was defined by Laszlo Lovasz to determine the Shannon capacity of the 5-cycle graph (Lovász, 1979) solving a problem that had been open in combinatorics for more than 20 years. This work subsequently inspired semidefinite approximation algorithms (Goemans & Williamson, 1995) and perfect graph theory (Berge, 2001). The Lovasz theta function requires the computation of a graph representation: for a given undirected graph $G(V, E)$ we would like to find unit norm vectors $\boldsymbol{v}_i$ where $i \in V$, such that non-adjacent vertices have orthogonal representations:

$$\boldsymbol{v}_i^T \boldsymbol{v}_j = 0, \quad \text{if } \{i, j\} \notin E.$$

Every graph has such a representation, if the dimension of the vectors $\boldsymbol{v}$ is not constrained. The Lovasz theta function searches for a graph representation that makes all these vectors fit in a small spherical cap. These representation vectors can be obtained by solving a semidefinite (SDP) program (see Gärtner & Matousek (2012) for a modern exposition).

Similarly, contrastive learning is a representation learning technique that yields impressive recent results (e.g. Chen et al. (2020b); He et al. (2020); Radford et al. (2021)). This training process aims to learn representations that have similar samples clustered together, while at the same time pulling different ones apart. This can be done in either an unsupervised fashion (i.e. without labels) or in a supervised way (Khosla et al., 2020). Contrastive learning approaches typically consider similarity between elements to be binary - two samples are similar (positive) or different (negative). However, it is natural for some problems to consider variability in similarity: Images of cats are closer to dogs compared to airplanes, and this insight can benefit representation learning.

**Our Contributions:** We establish a connection between contrastive learning and the Lovasz theta function. Specifically, we prove that the minimizers of the InfoNCE loss in the single positive case are the same (up to rotations) with those of the Lovasz theta optimum graph representation using an empty similarity graph.

Using this connection, we generalize contrastive learning using Lovasz theta on *weighted* graphs (Johansson et al., 2015). We define the Lovasz theta contrastive loss which leverages a weighted graph representing similarities between samples in each batch. Our loss is a generalization of the regular contrastive loss, since if positive examples are transformations of one sample and transformations of other images are used as negative examples (so the underlying graph corresponds to the empty one), we retrieve the regular contrastive loss. This way, any image similarity metric can be used to strengthen contrastive learning. For unsupervised contrastive learning, we show that our method can yield a benefit of up to $4\%$ over SimCLR for CIFAR100 using a pre-trained CLIP image

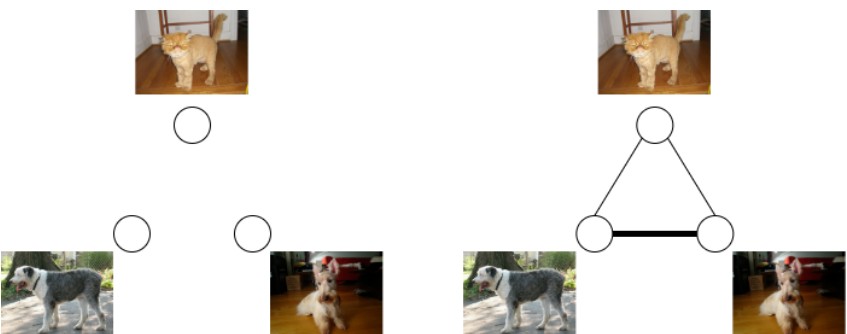

Figure 1: **Key idea of our method.** In this figure we can see how our proposed method works, with respect to the similarity graph between the classes. In the case of regular supervised contrastive learning (on the left), the similarity graph considered is just the empty graph (no class is similar to another). On the right, we see our proposed method. While different, the classes of dogs are similar to each other and different from cats, so the graph considered by our method reflects that (edge boldness reflects weight magnitude).

encoder to obtain similarities. For supervised contrastive learning (i.e. if class structure is used) our method yields a 1% benefit over supervised contrastive learning in CIFAR100 (Khosla et al., 2020).

The key idea of our method can be seen in Figure 1, where we see that we want to connect two samples more strongly if they are semantically related. There exists a body of work regarding graph representation learning Chen et al. (2020a), which learns functions on a given graph so that the distance between the representations is preserved. While related, our work differs from this, since our contrastive learning loss does not seek to explicitly retain the structure of the graph, but rather use it to guide how aligned or unaligned the representations of some samples must be.

In the rest of this work, we first examine the relationship between the Lovasz theta function and the InfoNCE loss. We then extend the latter via intuition derived from a weighted version of the Lovasz theta. Finally, we empirically evaluate our proposed loss, using several ways to model sample similarity.

## 2 RELATED WORK

### 2.1 LOVASZ THETA FUNCTION

The Lovasz theta function (Lovász, 1979) is a quantity which has been used to approximate the chromatic number of a graph. This is done by obtaining a representation for each vertex, which is clustered around a certain "handle" vector. One of the most important aspects of this function is that it is easily computable by solving an SDP (Gärtner & Matousek, 2012), despite the fact that the chromatic number it is used to approximate is very difficult to calculate in general.

### 2.2 SELF-SUPERVISED CONTRASTIVE LEARNING

There has been a flurry of activity in self-supervision, e.g. (Chen et al., 2020b; He et al., 2020; Oord et al., 2018; Chen & He, 2021; Wu et al., 2018). The main goal is to learn general representations that are good for a variety of downstream tasks. Ideally, we want to learn representations that are more general than those obtained by training classifiers. Wu et al. (2018) improved classification accuracy on ImageNet by a large margin over the state of the art. This was further simplified and improved by Chen et al. (2020b) by emphasizing the importance of data augmentations. More recent works (Oord et al., 2018; He et al., 2020; Chen & He, 2021) extend this by maximizing mutual information, adding a momentum encoder or a stop-gradient to one copy of their network, respectively. There has also been work on obtaining features at multiple granularities of an image to improve the learnt representations Zhou et al. (2022). Crucially, these works rely on large datasets of unlabeled data to learn quality representations that can then be applied to supervised learning tasks.

Recently, there has been work on mining hard negative samples for contrastive learning (Robinson et al., 2021). This line of work considers negative samples that are similar to a given sample as hard negatives, and uses those to perform contrastive learning. While related to our work, this approach is orthogonal, since it does not assume that the similarity between samples is a semantic relationship that we want to preserve.

Finally, a pair of works related to ours are those of HaoChen et al. (2021) and Shen et al. (2022). These works formulate the notion of a relationship graph between the given samples, by assuming that two samples are connected via an edge with weight equal to the probability that they correspond to views of the same sample. This is a natural measure of similarity between samples. In our work, we shall explicitly use these similarities to perform contrastive learning.

## 2.3 SUPERVISED CONTRASTIVE LEARNING

Along this line, Khosla et al. (2020) argued that when available we could leverage label information to learn good representations without large amounts of unlabeled data. They use a contrastive loss to pull together representations of samples belonging to the same class and push them apart from those of other samples. Training under this supervised contrastive loss shows an improvement on the classification accuracy on ImageNet compared to the cross-entropy loss, without any extra data.

### 2.3.1 MULTI-LABEL CONTRASTIVE LEARNING

Several works such as Radford et al. (2021) and Cui et al. (2017) use contrastive learning along with supervisory multi-labels such as tags to learn good representations. In particular, CLIP (Radford et al., 2021) maximizes the alignment between representations of an image and those of its captions.

The most closely related to our work is the recent paper Zhang et al. (2022) that leverages similarity gleaned from the multi-labels of the data. They define class similarity by using the hierarchical information which can be derived by the multi-label of the sample. They incorporate class similarity information in the contrastive loss by adding a penalty factor for the pairs of images from different classes. The penalty term pushes together representations for images that have similar labels higher in the hierarchy of the multi-label. Thus, they define class similarity by the level at which the labels are similar in the multi-label hierarchy. Note that if the labels are only of a single level, then this loss reduces to regular supervised contrastive loss.

In contrast to this work, our loss formulation is derived from a principled theoretical connection between contrastive loss and the Lovasz theta function of a graph. We define a new generalization of the contrastive loss that is derived from the Lovasz theta function. Moreover, we can easily incorporate similarities between samples into our setting, either via class similarities as in Zhang et al. (2022), or via directly assigning similarity to samples.

## 3 THE LOVASZ THETA FUNCTION

We start with the classic formulation for calculating the Lovasz theta for a graph.

**Definition 3.1** (Lovasz Theta of a Graph). Let $G = (V, E)$ be a given (unweighted) graph. Then we define as Lovasz theta of this graph, denoted as $\theta(G)$, the optimal value of the following minimization problem, for $N \leq d$:

$$
\begin{aligned}
\underset{\boldsymbol{u}_1,\ldots,\boldsymbol{u}_N,\boldsymbol{c}\in\mathbb{R}^d}{\text{minimize}} \quad & \max_{i=1,\ldots,N} \tfrac{1}{(\boldsymbol{c}^T\boldsymbol{u}_i)^2}, \\
\text{subject to} \quad & \|\boldsymbol{u}_1\|^2 = \|\boldsymbol{u}_2\|^2 = \cdots = \|\boldsymbol{u}_N\|^2 = \|\boldsymbol{c}\|^2 = 1, \\
& \boldsymbol{u}_i^T\boldsymbol{u}_j = 0, \ \forall (i,j) \notin E.
\end{aligned}
\tag{1}
$$

Essentially, the solution to this problem is a set of vectors $\boldsymbol{u}_i$ which are a) aligned with the "handle" vector $\boldsymbol{c}$ and b) are orthogonal to each other when the corresponding vertices of the graph are not connected. This problem is closely related to the chromatic number of the complementary graph. Indeed, we can think of the representations as being the "colors" assigned to each vertex, with the colors being different when the two vertices are connected in the complementary graph.

In the following, we shall use the Delsarte formulation of the Lovasz theta problem (Johansson et al., 2015; Schrijver, 1979):

$$\begin{aligned}
&\underset{\boldsymbol{u}_1,\ldots,\boldsymbol{u}_N,\boldsymbol{c}\in\mathbb{R}^d}{\text{minimize}} && \max_{i=1,\ldots,N} \frac{1}{(\boldsymbol{c}^T\boldsymbol{u}_i)^2}, \\
&\text{subject to} && \|\boldsymbol{u}_1\|^2 = \|\boldsymbol{u}_2\|^2 = \cdots = \|\boldsymbol{u}_N\|^2 = \|\boldsymbol{c}\|^2 = 1, \\
& && \boldsymbol{u}_i^T\boldsymbol{u}_j \leq 0, \ \forall (i,j) \notin E.
\end{aligned} \tag{2}$$

The reason for this is that this formulation is more amenable to an extension on weighted graphs, as we shall see in the next section.

One critical property of the Lovasz theta function is that, while it is used to approximate chromatic numbers on graphs (which are known to be NP-complete to calculate in general), it is actually easy to calculate. This is due to the following formulation of equation 2:

**Theorem 3.1** (Gärtner & Matousek 2012). *For $N \leq d$, the optimization problem in equation 2 can be rewritten in the following form:*

$$\begin{aligned}
&\text{minimize} && t, \\
&\text{subject to} && \boldsymbol{v}_i^T\boldsymbol{v}_j \leq t, \ \forall (i,j) \notin E, \\
& && \|\boldsymbol{v}_i\|^2 = 1, \ \forall i.
\end{aligned} \tag{3}$$

The above problem can be converted into a convex SDP by setting $\mathbf{Y} = \left[\boldsymbol{v}_i^T\boldsymbol{v}_j\right]_{i,j}$ (so we have $\mathbf{Y} \succeq 0$). Thus, the above problem tells us that for each vertex in the graph we want to find a unit representation, such that the vertices which are not connected by an edge have dissimilar representations (in other words, their inner product is minimized).

## 4 LOVASZ THETA AND CONTRASTIVE LEARNING

### 4.1 REGULAR CONTRASTIVE LEARNING

One of the most well-known losses used for contrastive learning is the InfoNCE loss (Chen et al., 2020b; Wu et al., 2018; Henaff, 2020; Van den Oord et al., 2018):

$$\mathcal{L}_{InfoNCE} = -\sum_{i=1}^{N} \log \frac{\exp(\boldsymbol{v}_+^T\boldsymbol{v}_i/\tau)}{\sum_{j\neq i}\exp(\boldsymbol{v}_j^T\boldsymbol{v}_i/\tau)}, \tag{4}$$

where $\boldsymbol{v}_+$ is the positive sample with respect to $\boldsymbol{v}_i$, and $\tau$ is a temperature parameter. For the theoretical analysis, we shall make the following simplifying assumptions:

- The representation vectors $\boldsymbol{v}_i \in \mathbb{R}^d$ are all unit norm. This constraint is often enforced in practice, with the representations produced by the network in question being normalized at the end. Moreover, we assume that $N \leq d$, or that the number of vectors is at most equal to their dimensionality. We refer the reader to Appendix A for further discussion.
- The only positive sample with respect to $\boldsymbol{v}_i$ is itself (single positive case). This nullifies the positive term (which serves for alignment between views of the same object).

Under these assumptions, we can reformulate our loss as follows:

$$\mathcal{L}_{InfoNCE} = \text{const} + \tau \sum_{i=1}^{N} \log \left( \sum_{j\neq i} \exp(\boldsymbol{v}_j^T\boldsymbol{v}_i/\tau) \right), \tag{5}$$

which we aim to minimize under the constraint $\|\boldsymbol{v}_i\|_2^2 = 1$, for all $i$. Note that we have multiplied the loss by $\tau > 0$, which does not affect the minimization. This loss is directly related to the Lovasz theta problem, as can be seen by the following theorem:

**Theorem 4.1** (Equivalence with Lovasz theta). *For $N \leq d$, the Delsarte formulation of the Lovasz theta problem:*

$$\begin{aligned}
&\text{minimize} && t, \\
&\text{subject to} && \|\boldsymbol{v}_i\|^2 = 1, \ \forall i, \\
& && \boldsymbol{v}_i^T\boldsymbol{v}_j \leq t, \ \forall i \neq j,
\end{aligned} \tag{6}$$

*and the minimization of the second term of the InfoNCE loss:*

$$
\begin{aligned}
\text{minimize} \quad & \tau \sum_{i=1}^{n} \log \sum_{j \neq i} \exp\left(\frac{\boldsymbol{v}_i^T \boldsymbol{v}_j}{\tau}\right), \\
\text{subject to} \quad & \|\boldsymbol{v}_i\|^2 = 1, \ \forall i,
\end{aligned}
\tag{7}
$$

*attain their minima at the same values of the matrix $\mathbf{Y} = \mathbf{V}^T \mathbf{V}$, where $\mathbf{V}$ is the matrix which has the vectors $\boldsymbol{v}_i$ as columns (so they have the same minimizers up to rotations).*

We defer the proof of this theorem to the Appendix. Intuitively, the connection between the two problems can be shown by bounding the second term of the InfoNCE loss and then considering the two problems as minimization over the inner products. It can then be shown that the two problems achieve the same value for the same inner products. This theorem directly links the novel field of contrastive learning with a classical problem in graph theory, thus providing a novel direction in the theoretical underpinnings of contrastive learning.

We can also consider the case of the Supervised Contrastive Loss (Khosla et al., 2020). In this setting, the positive samples for each image are not only its different versions, but rather all images belonging to the same class. Formally, we can write this loss as follows:

$$
L_{SupCon} = -\frac{1}{\tau} \sum_{i=1}^{N} \frac{1}{|P(i)|} \sum_{p \in P(i)} \boldsymbol{v}_p^T \boldsymbol{v}_i + \sum_{i=1}^{N} \log \left( \sum_{j \in N(i)} \exp(\boldsymbol{v}_j^T \boldsymbol{v}_i / \tau) \right),
\tag{8}
$$

where $P(i) = \{j : y_i = y_j\}$ and $N(i) = \{j : y_i \neq y_j\}$ the sets of positive and negative samples.

Our theorems above only regard the negative samples of the InfoNCE loss. While there are works which employ techniques like a slowly moving target (Grill et al., 2020) or a stop gradient on one part of the network (Chen & He, 2021) to avoid the use of negatives, they play a major role in the convergence of the simple formulation of the InfoNCE loss, encouraging uniformity across representations (Wang & Isola, 2020). By demonstrating this link of the negative samples with the Lovasz Theta, we can show an extension of the InfoNCE loss which can leverage this theoretical connection and use sample similarities within a batch, providing us with an alternative use for these samples.

## 4.2 EXTENSION TO SIMILARITY GRAPHS

Using the above formulation of the problem, we can leverage the graph theoretical aspect of the Lovasz theta in order to incorporate similarity information into our contrastive problem. Let us assume that we have a weighted graph $G(V, W)$, with each vertex corresponding to a sample. In this graph, the weights $w_{ij}$ correspond to the similarity between the samples $i$ and $j$. As such, we have the condition that $w_{i,j} \in [0, 1]$ (in the case where this matrix of weights $W$ becomes the identity matrix, then this formulation reduces to the regular contrastive loss). In this case, we use a weighted version of the Lovasz theta problem, where we also relax the equality conditions to inequalities (Johansson et al., 2015). This is formulated as:

$$
\begin{aligned}
\underset{\boldsymbol{u}_1,\ldots,\boldsymbol{u}_N,\boldsymbol{c}\in\mathbb{R}^d}{\text{minimize}} \quad & \max_{i=1,\ldots,N} \frac{1}{(\boldsymbol{c}^T \boldsymbol{u}_i)^2}, \\
\text{subject to} \quad & \|\boldsymbol{u}_1\|^2 = \|\boldsymbol{u}_2\|^2 = \cdots = \|\boldsymbol{u}_N\|^2 = \|\boldsymbol{c}\|^2 = 1, \\
& \boldsymbol{u}_i^T \boldsymbol{u}_j \leq w_{ij}, \ \forall i \neq j,
\end{aligned}
\tag{9}
$$

where $w_{ij}$ in this case is the similarity between samples $i$ and $j$. Indeed, if the weight approaches 0, the corresponding constraints become the same as in the Lovasz theta problem. If the weight approaches 1, then these samples are perfectly connected, and the corresponding constraint on their inner product becomes trivial. Any value $w_{ij} \in (0, 1)$ is a degree of similarity – the less similar the sample are, the more spread apart the representations need to be to satisfy the stricter upper bound.

**Lemma 4.1.** *For $N \leq d$, the problem in equation 9 is equivalent to:*

$$
\begin{aligned}
\underset{\boldsymbol{v}_1,\ldots,\boldsymbol{v}_N\in\mathbb{R}^d}{\text{minimize}} \quad & t, \\
\text{subject to} \quad & \|\boldsymbol{v}_1\|^2 = \|\boldsymbol{v}_2\|^2 = \cdots = \|\boldsymbol{v}_N\|^2 = 1, \\
& \boldsymbol{v}_i^T \boldsymbol{v}_j \leq w_{ij} + (1 - w_{ij})t, \ \forall i \neq j.
\end{aligned}
\tag{10}
$$

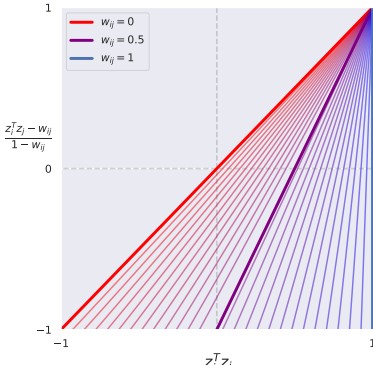

Figure 2: **Illustration of the effect of similarity terms $w_{ij}$ in the Lovasz contrastive loss.** The x-axis denotes the dot product between two features, and the y-axis represents the dot-product after the similarity terms are applied. For completely dissimilar samples, $w_{ij} = 0$ and this reduces to the contrastive supervised loss. For partially similar samples, the contribution to the negative-sample term of the contrastive loss is decreased. As the similarity increases to 1, this inner product vanishes from the Log-Sum-Exp of the uniformity term of the contrastive loss (the second term).

Using the Lovasz theta on a weighted graph, we can create the following form of contrastive loss:

$$L_{LovaszCon} = -\sum_{i=1}^{N} \frac{1}{|P(i)|} \sum_{p \in P(i)} \boldsymbol{v}_p^T \boldsymbol{v}_i + \tau \sum_{i=1}^{N} \log \left( \sum_{j \neq i} \exp \left( \frac{\boldsymbol{v}_j^T \boldsymbol{v}_i - w_{ij}}{\tau(1 - w_{ij})} \right) \right). \qquad (11)$$

Compared to $L_{SupCon}$, we have multiplied the loss function by $\tau > 0$. For finite $\tau > 0$, this makes no difference in the minimization, but for $\tau \to 0$, this allows us to state the following:

**Theorem 4.2.** *Consider the weighted Lovasz theta problem in equation 10 and the minimization of the second term of the Lovasz theta contrastive loss:*

$$\begin{array}{ll} \underset{\boldsymbol{v}_1,\ldots,\boldsymbol{v}_N \in \mathbb{R}^d}{\text{minimize}} & \tau \sum_{i=1}^{N} \log \left( \sum_{j=1}^{N} \exp \left( \frac{\boldsymbol{v}_j^T \boldsymbol{v}_i - w_{ij}}{\tau(1 - w_{ij})} \right) \right), \\ \text{subject to} & \|\boldsymbol{v}_1\|^2 = \|\boldsymbol{v}_1\|^2 = \|\boldsymbol{v}_2\|^2 = \cdots = \|\boldsymbol{v}_N\|^2 = 1. \end{array} \qquad (12)$$

*Minimizing the limit of the objective of the second problem as $\tau \to 0$ is a relaxation of the first problem, if $w_{ij} < 1$ for $i \neq j$.*

We defer all proofs to the Appendix. This connection is what allows us to leverage equation 3 to transform the loss we use into equation 11. In the unsupervised case, the first term consists of different views of the same sample. In any case, we make use of the positive term in our final loss.

We can make a few more remarks regarding the term $\frac{\boldsymbol{v}_j^T \boldsymbol{v}_i - w_{ij}}{\tau(1 - w_{ij})}$ in the exponent. As $w_{ij}$ approaches 0 (or in other words, as the samples become less and less related), the fraction within the exponents in the above expression becomes closer to the one present in the regular supervised contrastive loss. As $w_{ij}$ approaches 1 (so the two samples become more and more similar), if $\boldsymbol{v}_i \neq \boldsymbol{v}_j$ (which is satisfied assuming that the representations are not degenerate), the denominator goes to 0, but the numerator also becomes negative, making the term after the exponentiation negligibly small. In this case, the constraint in equation 10 becomes trivial (since the vectors have unit norm).

The two points above demonstrate that our technique is a natural extension of contrastive learning. Indeed, the latter is a special case of the former, when the similarities are considered to be the identity matrix. We can also graphically see the effect of $w_{ij}$ in each of the terms of the Log-Sum-Exp function. This effect is visualized in Figure 2, where we can see that $w_{ij}$ varies the slope of the term inside the exponent. The higher $w_{ij}$ is, the less this term is penalized if the inner product is large (so the negative terms are penalized less if the samples are more similar).

One major design choice in the above formulation is that of the similarity graph that we use. We identify two major approaches that can be used to derive the similarity matrix.

Table 1: **Summary of our results on CIFAR100, unsupervised case.** We compare our method with SimCLR, using the implementation provided by the authors of Khosla et al. (2020), and MoCo-v2, using the official implementation by the authors of He et al. (2020). The code for the baselines was run with the instructions provided by the authors' github pages, and was ran for 300 and 1000 epochs. We also experimented with two ways to derive our similarity matrix from CLIP.

|  | MoCo-v2 | SimCLR | Ours (cutoff) | Ours |
|---|---|---|---|---|
| ResNet-50, 300 epochs | 58.25 | 64.42 | $67.52 \pm 0.79$ | $\mathbf{68.14 \pm 0.46}$ |
| ResNet-18, 300 epochs | 51.96 | 58.64 | $60.39 \pm 1.01$ | $\mathbf{60.70 \pm 0.73}$ |
| ResNet-50, 1000 epochs | 64.43 | 68.27 | $70.22 \pm 0.08$ | $\mathbf{70.30 \pm 0.40}$ |
| ResNet-18, 1000 epochs | 58.94 | 63.41 | $64.88 \pm 0.12$ | $\mathbf{65.24 \pm 0.19}$ |

- **Supervised case.** If we assume that we have access to the labels of the samples during contrastive training, then we can derive the similarities between samples via the similarities between their respective classes. A simple way to do this is the following:

  1. We obtain a confusion matrix $\mathbf{C} = [c_{ij}]_{i,j=1,\ldots,N_{classes}}$ from a pretrained classifier.
  2. We normalize the confusion matrix across its rows (so that they all sum to 1).
  3. We set $\mathbf{C}' = [c'_{kl}]_{k,l=1,\ldots,N_{classes}}$, where $c'_{kl} = \frac{1}{2}(c_{kl} + c_{lk})$ if $k \neq l$ and $c'_{kl} = 1$ otherwise.

  This matrix $\mathbf{C}'$ is our class similarity matrix, and given two samples $x_i$ and $x_j$, with corresponding classes $y_i$ and $y_j$, we can define their similarity as $w_{ij} = c'_{y_i y_j}$.

  Moreover, we use domain knowledge for the problem, and derive our similarity matrix in an alternative fashion. Given a hierarchical structure for the classes of the problem, we can derive similarity by considering classes that belong in the same hierarchical superclass as being similar. We examine the above choices in the experimental section.

- **Unsupervised case.** If we assume that we do not have access to the labels of our samples during the contrastive training process (or that only part of them is labeled), then we could also leverage a pre-existing model to derive sample similarities directly. For example, we could use a pretrained image encoder such as CLIP (Radford et al., 2021), in order to derive a unit norm embedding $v_i$ for each sample $x_i$, and define the similarity between the two samples as $w_{ij} = v_i^T v_j$. While in our experiments we didn't observe negative similarities arising from this we also experiment by explicitly setting a cutoff in our similarities (setting any negative ones to 0), to make this consistent with the rest of the similarity matrices.

We refer the reader to the Appendix for further analysis. We also note the connection of this formulation with the work of Shen et al. (2022). where the authors cast contrastive learning as a spectral graph theory problem on the graph linking the views of the various samples. This graph has edges between views, with weight equal to the probability that the two views belong to the same sample.

## 5 EXPERIMENTS

We now examine the above formulation experimentally, by training models using the proposed form of contrastive loss. To do this, we first train our model for a set amount of epochs on the given dataset, using the Lovasz theta paradigm we described above. The result of this training is a model which produces a set of good representations for the images in this dataset. We then freeze the learned representations and learn a linear model on top of them. This linear classifier essentially considers the features of an image to be the outputs of the previous contrastively trained model. We evaluate the quality of our representations using the performance of this linear classifier as a metric, as in Khosla et al. (2020), by freezing the model learned during the contrastive procedure, and training a linear classifier on top of the frozen representations for a small number of epochs. We also employ a technique commonly used in contrastive learning frameworks, where the contrastive loss is applied not on the representations $v$, but rather on the output of a small head using these representations as input, namely on $z = g_{head}(v)$ (Khosla et al., 2020; Chen et al., 2020b). After contrastive training is performed, this head is discarded, and the representations given to it as input

Table 2: **Summary of our results on CIFAR100, supervised case.** We can see that our method both training with only crossentropy (CE) and the supervised contrastive baseline (SupCon). Our method is implemented by constructing the similarity matrix via the confusion matrix, or via the superclasses in CIFAR100. The numbers with * are taken from Khosla et al. (2020).

|  | CE | SupCon | Ours (Confusion Matrix) | Ours (Superclass) |
|---|---|---|---|---|
| ResNet-50 | 75.3* | 76.5* | $77.15 \pm 0.11$ | $\mathbf{77.60 \pm 0.30}$ |
| ResNet-34 | 74.98 | 75.81 | $76.06 \pm 0.29$ | $\mathbf{76.55 \pm 0.40}$ |
| ResNet-18 | 72.67 | 73.78 | $74.63 \pm 0.18$ | $\mathbf{74.91 \pm 0.15}$ |

Table 3: **Results on our ImageNet-100 experiment.** We can see that our method yields slightly better accuracy over both classical Cross Entropy loss and Supervised Contrastive Learning.

|  | CE | SupCon | Ours |
|---|---|---|---|
| ResNet-18 | 81.08 | 81.20 | **81.58** |
| ResNet-34 | 82.52 | 82.66 | **82.90** |

are used. In all of our comparisons, we examine ResNets of different sizes as the models trained in the contrastive fashion. Further experimental details can be found in the Appendix.

**Unsupervised Similarity Matrix.** In this setting, we derive our similarity matrix using the unsupervised approach mentioned above. We use CLIP to derive the cosine similarities between samples, and employ them in order to perform Lovasz theta contrastive training on our method. The results of our method can be seen in Table 1. This evaluation is similar to the one performed in Khosla et al. (2020) in that the feature extractor is trained from scratch using the CIFAR100 dataset, and not using a transfer learning approach as is done in other works (Chen et al., 2020b; Grill et al., 2020).

We can see that our method outperforms regular MoCo-v2 and SimCLR pretraining on CIFAR100. This shows that using our similarity matrix, we can take advantage of sample similarities in the case where labels for each of the samples are not immediately available during contrastive pretraining. We refer the reader to Appendix C.1 for further discussion on the quality of CLIP representations.

**Supervised Similarity Matrix.** In this setting, we consider the comparison of our method to two supervised baselines. The first one is simple crossentropy loss (CE) where the model is trained in a supervised fashion. The second one is that of supervised contrastive learning (SupCon) (Khosla et al., 2020). As mentioned above, this technique is a special case of our method, where each sample is considered perfectly similar only to itself (hence the similarity graph is only the identity matrix). As such, a direct comparison with it allows us to fully understand the benefits that the similarity matrix provides to our method. We use the evaluation process we described above, on CIFAR100 and ImageNet-100 (a subset of 100 classes of ImageNet (Tian et al., 2020)) using 3 different ResNet architectures. For CIFAR100, define the similarity matrix via either a) the confusion matrix, derived from a model trained with SupCon, or b) via the 20 superclasses defined over the 100 classes of CIFAR100 (Krizhevsky et al., 2009). The similarity between two classes belonging to the same superclass was considered as a hyperparameter, and was set equal to 0.5. For ImageNet-100, we derive the similarities using a pretrained ResNet-50 provided by Pytorch. We can see our results in Tables 2 and 3. From our results on CIFAR100 we can infer that the use of the similarity matrix is indeed helpful to the training process. Intuitively, relaxing the constraint on the negatives allows the loss to achieve a better representation overall. A similar conclusion can be obtained from our results on ImageNet-100, where again our method performs better than the baselines. We refer the reader to the Appendix for results on CIFAR10 as well as further experimental details.

**Ablation on Similarity Matrix.** As noted previously, a major design choice in our experiments is the method used to derive the similarity matrix, access to which is assumed by our method. Here, we test this choice on CIFAR100, in the supervised setting. Namely, we consider the following two

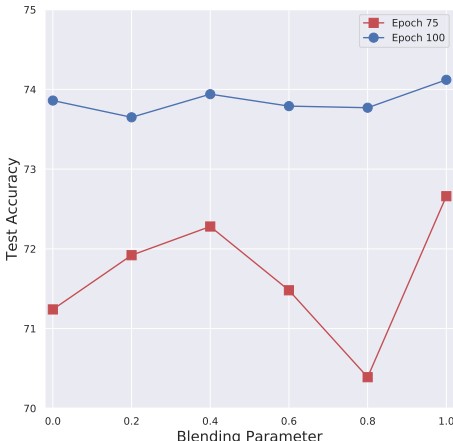

Figure 3: **Evolution of loss based on blending of similarity matrix.** Both plots are based on CIFAR100, with models pretrained for 75 and 100 epochs. The blending parameter varies from 0 (similarity matrix identity as in SupCon) to 1 (similarity matrix derived by the confusion matrix).

Table 4: **Ablation on the choice of the similarity matrix.** Our two proposed choices are retrieving the similarities via the superclasses of the problem, and the other is to examine the inner products of the vectors produced by the text representations of CLIP. The first approach is better, but the CLIP approach gets adequate results, without explicit domain information other than the text labels.

|  | Superclass Similarity | CLIP |
| --- | --- | --- |
| ResNet-50 | 77.60 | 76.25 |
| ResNet-34 | 76.55 | 75.19 |
| ResNet-18 | 74.91 | 73.56 |

methods to derive these similarities discussed above: using the superclass similarities provided for CIFAR100 and using the inner products of the CLIP text representations of the labels. Our results can be seen in Table 4. We can see that the superclass similarity approach achieves better accuracy in all 3 of the considered models. However, we should note here that retrieving the similarities through CLIP requires access to only the labels of the problem, and is thus easier to retrieve.

**Path from Contrastive Learning to Lovasz Theta Contrastive Learning.** A final experiment that we conduct is varying the similarity matrix between the confusion matrix and the identity. For the supervised setting, if $\mathbf{C}$ is our class similarity matrix, we set $\mathbf{C}' = \lambda\mathbf{C} + (1 - \lambda)\mathbf{I}$, as our new similarity matrix, where $\lambda \in [0, 1]$ is a blending parameter. This can show how our performance changes from $\lambda = 0$ (regular SupCon) to $\lambda = 1$ (our method). The results can be seen in Figure 3. We see that there is a subtle and non-monotonic, but overall increasing trend in performance as the similarity matrix moves closer to ours. As such, this blending can be tuned to improve the results.

## 6 CONCLUSIONS

We established a connection between the InfoNCE loss and the Lovasz theta of a graph. This allowed us to generalize contrastive learning using general similarity graphs. Our technique can use any method of measuring similarity between samples to create a problem-specific constrastive loss. Our experiments show benefits over regular contrastive learning, in both the supervised and the unsupervised case, using simple similarity metrics. This is natural since we use additional information provided in the similarity structure. The design decision of how to measure similarity between samples is central for our loss and opens an interesting direction for future research.

ETHICS STATEMENT

We do not consider our work to have any immediate ethical considerations.

REPRODUCIBILITY STATEMENT

The proofs for all of the theorems in the paper are included in the Appendix. Regarding the experiments, we have included details for hyperparameters in the Appendix, and we have also provided anonymized code as part of the supplementary material. This code will be moved to a public repository after the reviewing process.

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

APPENDIX

## A    DEFERRED THEOREMS & PROOFS

In this section, we include further theoretical results, including the proofs omitted from the main body of the paper.

**On the assumption $N \leq d$.**    The assumption on the number of vectors allows us to convert the Lovasz theta problem into a convex one. Indeed, without this assumption, converting the problem into an SDP by setting $\mathbf{Y} = \mathbf{V}^T\mathbf{V}$ would impose constraints on the rank of the matrix $\mathbf{Y}$ (specifically, $\text{rank}(\mathbf{Y}) \leq d$). Since we know that $\text{rank}(\mathbf{Y}) \leq N$ by construction, then having $N \leq d$ makes this rank constraint trivial. If we had these rank constraints, then the problem would be non-convex, given that the domain of the problem (a set of rank constrained matrices) is non-convex.

In practice, whether this constraint is satisfied is based on our choice for model dimensionality and batch size. It can however be circumvented, by regular optimization methods operating directly on the representations $\boldsymbol{v}_1, \ldots, \boldsymbol{v}_N$ (rather than their inner products). While the problem may no longer be convex, we can still apply first order optimization methods to find a good stationary point for the loss.

**Proof of Theorem 3.1.**    The following is an adaptation of the proof found in Gärtner & Matousek (2012).

We first show that the following problems:

$$
\begin{aligned}
&\underset{\boldsymbol{u}_1,\ldots,\boldsymbol{u}_N,\boldsymbol{c}\in\mathbb{R}^d}{\text{minimize}} && k, \\
&\text{subject to} && \|\boldsymbol{u}_1\|^2 = \|\boldsymbol{u}_2\|^2 = \cdots = \|\boldsymbol{u}_N\|^2 = \|\boldsymbol{c}\|^2 = 1, \\
& && \boldsymbol{u}_i^T\boldsymbol{u}_j \leq 0, \ \forall (i,j) \notin E, \\
& && \tfrac{1}{(\boldsymbol{c}^T\boldsymbol{u}_i)^2} \leq k, \ \forall i,
\end{aligned}
\tag{13}
$$

and:

$$
\begin{aligned}
&\underset{\boldsymbol{v}_1,\ldots,\boldsymbol{v}_N\in\mathbb{R}^d}{\text{minimize}} && k, \\
&\text{subject to} && \|\boldsymbol{v}_1\|^2 = \|\boldsymbol{v}_2\|^2 = \cdots = \|\boldsymbol{v}_N\|^2 = 1, \\
& && \boldsymbol{v}_i^T\boldsymbol{v}_j \leq -\tfrac{1}{k-1}, \ \forall (i,j) \notin E, \\
& && {\color{blue} k > 1}
\end{aligned}
\tag{14}
$$

are equivalent. To do this, we shall show that an optimal solution of the first problem can be converted to a feasible solution for the second one, and vice versa.

Given an optimal solution to the first problem $\boldsymbol{u}_1, \ldots, \boldsymbol{u}_N, \boldsymbol{c}$ with optimal value $k$, we can define the following set of vectors:

$$
\boldsymbol{z}_i = \frac{1}{\sqrt{k-1}}\left(\frac{\boldsymbol{u}_i}{\boldsymbol{c}^T\boldsymbol{u}_i} - \boldsymbol{c}\right).
\tag{15}
$$

{\color{blue} Note that the conditions of the first problem immediately imply that $k \geq 1$, as $0 \leq (\boldsymbol{c}^T\boldsymbol{u}_i)^2 \leq 1 \Rightarrow 1 \leq \frac{1}{(\boldsymbol{c}^T\boldsymbol{u}_i)^2} \leq k$. Thus, the optimal solution will have $k > 1$ - we can safely ignore the case $k = 1$, as this implies that all the vectors are aligned, which can only happen if all vertices are connected. In this case the second problem will also have no constraints to be satisfied other than $k > 1$ ($t < 0$), and with no constraints $t = -\frac{1}{k-1}$ approaches $-\infty$, which is consistent with $k = 1$.} For these vectors, the following hold:

- $\boldsymbol{z}_i^T\boldsymbol{z}_j = \frac{1}{k-1}\left(\frac{\boldsymbol{u}_i^T\boldsymbol{u}_j}{(\boldsymbol{c}^T\boldsymbol{u}_i)(\boldsymbol{c}^T\boldsymbol{u}_j)} - 1 - 1 + 1\right) \leq -\frac{1}{k-1}$.

- $\|\boldsymbol{z}_i\|^2 = \frac{1}{k-1}\left(\frac{\|\boldsymbol{u}_i\|^2}{(\boldsymbol{c}^T\boldsymbol{u}_i)^2} - 1 - 1 + 1\right) \leq \frac{1}{k-1}(k-1) = 1$.

Let us now define the matrix $\mathbf{Z}$, with columns the vectors $\boldsymbol{z}_i$. Thus, $\mathbf{Z}^T\mathbf{Z}$ is positive semidefinite (by construction) and has diagonal elements which are less than or equal to $1$. Thus, we can define the matrix $\mathbf{Y} = \mathbf{Z}^T\mathbf{Z} + \mathbf{D}$, where $\mathbf{D}$ is a diagonal matrix with non-negative elements, such that $y_{ii} = 1$. This matrix $\mathbf{Y}$ is also PSD, which means that we can write it as $\mathbf{Y} = \mathbf{V}^T\mathbf{V}$. The columns of $\mathbf{V}$

are precisely the vectors $\boldsymbol{v}_i$ that form a feasible solution of the second problem, since their inner products (off-diagonal elements of $\mathbf{Y}$) satisfy the required constraints, and their norms (diagonal elements of $\mathbf{Y}$) are equal to 1.

Now, given an optimal solution to the second problem, $\boldsymbol{v}_1, \ldots, \boldsymbol{v}_N$ with optimal value $k$, we can again define $\mathbf{Y} = \mathbf{V}^T \mathbf{V}$. Here we can make an argument that $\mathbf{Y}$ must have at least one eigenvalue equal to 0 (in other words, the vectors $\boldsymbol{v}_i$ must be linearly dependent). Indeed, if this was not the case, we would have $\mathbf{Y} \succ 0$, so $\mathbf{Y} - \epsilon \mathbf{I} \succeq 0$, where $\epsilon > 0$ sufficiently small. This means that the matrix $\mathbf{Y}' = \frac{1}{1-\epsilon}(\mathbf{Y} - \epsilon \mathbf{I})$ is a feasible solution to the second problem (note that the off diagonal elements of $\mathbf{Y}$ are negative, so multiplying them by a positive constant decreases them) which also has lower value for the objective function, which is a contradiction. Thus $\mathbf{Y}$ is singular, which also implies that there exists a unit vector $\boldsymbol{c} \in \mathbb{R}^d$ that has $\boldsymbol{c}^T \boldsymbol{v}_i = 0$, for all $i$. We can then define the following vectors:

$$\boldsymbol{u}_i = \frac{1}{\sqrt{k}}(\boldsymbol{c} + \boldsymbol{v}_i \sqrt{k-1}). \tag{16}$$

For these vectors, we have:

- $\boldsymbol{u}_i^T \boldsymbol{u}_j = \frac{1}{k}(1 + 0 + 0 + (k-1)\boldsymbol{v}_i^T \boldsymbol{v}_j) \leq 0.$
- $\|\boldsymbol{u}_i\|^2 = \frac{1}{k}(1 + 0 + 0 + k - 1) = 1.$
- $\boldsymbol{c}^T \boldsymbol{u}_i = \frac{1}{\sqrt{k}} \Rightarrow \frac{1}{(\boldsymbol{c}^T \boldsymbol{u}_i)^2} = k.$

This means that the vectors $\boldsymbol{u}_i$ along with $\boldsymbol{c}$ form a feasible solution for the first problem, with objective value $k$.

For the final step, we note that by setting $t = -\frac{1}{k-1}$, then by minimizing $k$ we also minimize $t$ (since the latter is an increasing function of $k$). This means that our second problem is equivalent to:

$$\begin{aligned} \underset{\boldsymbol{v}_1, \ldots, \boldsymbol{v}_N \in \mathbb{R}^d}{\text{minimize}} \quad & t, \\ \text{subject to} \quad & \|\boldsymbol{v}_1\|^2 = \|\boldsymbol{v}_2\|^2 = \cdots = \|\boldsymbol{v}_N\|^2 = 1, \\ & \boldsymbol{v}_i^T \boldsymbol{v}_j \leq t, \; \forall (i,j) \notin E \\ & t < 0 \end{aligned} \tag{17}$$

However, we note that the constraint $t < 0$ can be dropped from the problem. This is due to the fact that the rest of the constraints, for $N \leq d$, can be satisfied by the solution where all the $v_i$ are orthogonal, for example $v_i = e_i$, which would give $t = 0$. Using this solution as a base, we have $\mathbf{Y} = \mathbf{V}^T \mathbf{V} = \mathbf{I}$, where $\mathbf{V}$ is the matrix with $v_i$ as columns. We can then form $\mathbf{Y}' = (1 + \epsilon)\mathbf{I} - \epsilon \mathbf{1}$, where $\epsilon > 0$ arbitrarily small and $\mathbf{1}$ the all-ones matrix. This new matrix will have $y'_{ii} = 1$ and $y'_{ij} = -\epsilon$ for all $(i,j) \notin E$. Moreover, it will be positive definite, as $\mathbf{1}$ has $N-1$ eigenvalues equal to 0, and one of them equal to $N$, and so this implies that $\mathbf{Y}'$ has $N-1$ eigenvalues equal to $1 + \epsilon > 0$ and one of them equal to $1 + \epsilon - N\epsilon = 1 - (N-1)\epsilon > 0$ for $\epsilon$ arbitrarily small. Thus, we can find a feasible solution $v'_1, \ldots, v'_N$ such that $\mathbf{Y}' = \mathbf{V}'^T \mathbf{V}'$ and with $t = -\epsilon < 0$. Thus, the optimal solution must have $t < 0$, meaning that the constraint can be dropped. This completes our proof.

**Theorem A.1.** *Theorem 4.1 in the main paper. For $N \leq d$, the Delsarte formulation of the Lovasz theta problem:*

$$\begin{aligned} \text{minimize} \quad & t, \\ \text{subject to} \quad & \|\boldsymbol{v}_i\|^2 = 1, \; \forall i, \\ & \boldsymbol{v}_i^T \boldsymbol{v}_j \leq t, \; \forall i \neq j, \end{aligned} \tag{18}$$

*and the minimization of the second term of the InfoNCE loss:*

$$\begin{aligned} \text{minimize} \quad & \tau \sum_{i=1}^n \log \sum_{j \neq i} \exp\left(\frac{\boldsymbol{v}_i^T \boldsymbol{v}_j}{\tau}\right), \\ \text{subject to} \quad & \|\boldsymbol{v}_i\|^2 = 1, \; \forall i, \end{aligned} \tag{19}$$

*attain their minima at the same values of the matrix $\mathbf{Y} = \mathbf{V}^T \mathbf{V}$, where $\mathbf{V}$ is the matrix which has the vectors $\boldsymbol{v}_i$ as columns.*

*Proof.* Note that by setting $\mathbf{Y} = \mathbf{V}^T\mathbf{V}$, the first problem is converted into:

$$
\begin{aligned}
\text{minimize} \quad & t, \\
\text{subject to} \quad & y_{ii} = 1, \; \forall i, \\
& y_{ij} \leq t, \; \forall i \neq j, \\
& \mathbf{Y} \succeq 0,
\end{aligned}
\tag{20}
$$

while the second problem is converted into:

$$
\begin{aligned}
\text{minimize} \quad & \tau \sum_{i=1}^{n} \log \sum_{j \neq i} \exp\left(\frac{y_{ij}}{\tau}\right), \\
\text{subject to} \quad & y_{ii} = 1, \; \forall i, \\
& \mathbf{Y} \succeq 0.
\end{aligned}
\tag{21}
$$

Let us assume that we have a feasible solution $\mathbf{Y}$ for the second problem. Let $t = \frac{1}{n(n-1)} \sum_{i,j:i\neq j} y_{ij}$ (the average of the non-diagonal elements of $\mathbf{Y}$). Due to the convexity of the exponential function and the logarithm being an increasing function, we can apply Jensen's inequality in each of the sums inside the logarithms for each given $i$, thus giving us:

$$
\begin{aligned}
\tau \sum_{i=1}^{n} \log \sum_{j \neq i} \exp\left(\frac{y_{ij}}{\tau}\right) &\geq \tau \sum_{i=1}^{n} \log \left( (n-1) \exp\left( \frac{\frac{1}{n-1}\sum_{j\neq i} y_{ij}}{\tau} \right) \right) \\
&= \tau n \log(n-1) + \frac{1}{n-1} \sum_{i=1}^{n} \sum_{j \neq i} y_{ij} \\
&= \tau n \log(n-1) + nt.
\end{aligned}
\tag{22}
$$

The final step is precisely the value of our objective function, when all $y_{ij}$, $i \neq j$ are equal to $t$. This means that, if we replace our solution with the average of the non-diagonal elements, then we will always decrease the value of the objective. Note that the new solution will still be feasible. Indeed, since the original matrix $\mathbf{Y}$ is PSD, we have:

$$
\mathbf{1}^T\mathbf{Y}\mathbf{1} = n + n(n-1)t \geq 0 \Rightarrow t \geq -\frac{1}{n-1}.
\tag{23}
$$

The new matrix we will use is:

$$
\mathbf{Y}' = (1-t)\mathbf{I} + t\mathbf{1},
\tag{24}
$$

(so the diagonal elements are 1, and the non-diagonal ones are $t$). This is a matrix with rank-1 difference from the identity, and it has $n-1$ eigenvalues equal to $1-t \geq 0$ (since $t$ is the average of inner products of vectors with unit norm), and one eigenvalue equal to $1-t+tn = 1+(n-1)t \geq 0$ (due to the above). Thus the new matrix is also a feasible solution, so it is always optimal to have $y_{ij} = t$.

Thus, we can rewrite our problem as follows:

$$
\begin{aligned}
\text{minimize} \quad & \tau \sum_{i=1}^{n} \log \sum_{j \neq i} \exp\left(\frac{y_{ij}}{\tau}\right), \\
\text{subject to} \quad & y_{ii} = 1, \; \forall i, \\
& y_{ij} = t, \; i \neq j, \\
& \mathbf{Y} \succeq 0.
\end{aligned}
\tag{25}
$$

Given the condition for $y_{ij}$, we can rewrite our objective function as:

$$
\tau \sum_{i=1}^{n} \log \left( n(n-1) \exp\left(\frac{t}{\tau}\right) \right) = \tau n \log\left(n(n-1)\right) + tn.
\tag{26}
$$

Thus, we can easily see that this problem has the same minimizing matrix $\mathbf{Y}$ as:

$$
\begin{aligned}
\text{minimize} \quad & t, \\
\text{subject to} \quad & y_{ii} = 1, \; \forall i, \\
& y_{ij} = t, \; i \neq j, \\
& \mathbf{Y} \succeq 0.
\end{aligned}
\tag{27}
$$

Finally, we need to argue that this problem has the same optimal solution as equation 20 (or that, in other words, having the constraints $y_{ij} = t$ be $y_{ij} \leq t$ does not change the optimal solution).

This can be easily shown using the exact same argument as above - if we assume that there exists an element $y_{ij} < t$ in the optimal solution, then we can replace the non-diagonal elements of $\mathbf{Y}$ with their average $\bar{y} < t$, giving us a feasible solution $\mathbf{Y}' \succeq 0, t' = \bar{y} < t$, which is impossible. Thus, the problem in equation 20 has the same minimizer matrix $\mathbf{Y}$ as that in equation 27. This completes the proof, as having the same matrix $\mathbf{Y} = \mathbf{V}^T\mathbf{V}$ and having the vectors all be equal in norm means that the vectors chosen are unique up to rotations. □

**Lemma A.1.** *The following holds:*

$$\lim_{\tau \to 0^+} \tau \log \sum_{i=1}^{n} \exp \frac{z_i}{\tau} = \max_{i=1}^{n} z_i. \tag{28}$$

*Proof.* We have:

$$\sum_{i=1}^{n} \exp \frac{z_i}{\tau} \geq \exp\left(\frac{1}{\tau} \max_{i=1}^{n} z_i\right) \Rightarrow \max_{i=1}^{n} z_i \leq \tau \log \sum_{i=1}^{n} \exp \frac{z_i}{\tau}, \tag{29}$$

as well as:

$$\tau \log \sum_{i=1}^{n} \exp \frac{z_i}{\tau} \leq \tau \log\left(n \exp\left(\frac{1}{\tau} \max_{i=1}^{n} z_i\right)\right) = \tau \log n + \max_{i=1}^{n} z_i. \tag{30}$$

Thus, we get:

$$\max_{i=1}^{n} z_i \leq \tau \log \sum_{i=1}^{n} \exp \frac{z_i}{\tau} \leq \tau \log n + \max_{i=1}^{n} z_i, \tag{31}$$

and taking the limit as $\tau \to 0$ gives us the desired result.

We note here that the convergence is **uniform**: for a given $\tau$, the difference between $\tau \log \sum_{i=1}^{n} \exp \frac{z_i}{\tau}$ and $\max_{i=1}^{n} z_i$ is upper bounded by $\tau \log n$, which is independent of $z_1, \ldots, z_n$. □

**Lemma A.2** (Lemma 4.1 in the main paper). *For $N \leq d$, the weighted Lovasz Theta problem can be rewritten as in equation 33. In other words, the following formulations of the weighted Lovasz theta problem are equivalent:*

$$\begin{array}{ll}
\underset{\boldsymbol{u}_1,\ldots,\boldsymbol{u}_N,\boldsymbol{c}\in\mathbb{R}^d}{\text{minimize}} & \max_{i=1,\ldots,N} \frac{1}{(\boldsymbol{c}^T\boldsymbol{u}_i)^2}, \\
\text{subject to} & \|\boldsymbol{u}_1\|^2 = \|\boldsymbol{u}_2\|^2 = \cdots = \|\boldsymbol{u}_N\|^2 = \|\boldsymbol{c}\|^2 = 1, \\
& \boldsymbol{u}_i^T\boldsymbol{u}_j \leq w_{ij},
\end{array} \tag{32}$$

*and:*

$$\begin{array}{ll}
\underset{\boldsymbol{v}_1,\ldots,\boldsymbol{v}_N\in\mathbb{R}^d}{\text{minimize}} & t, \\
\text{subject to} & \|\boldsymbol{v}_1\|^2 = \|\boldsymbol{v}_2\|^2 = \cdots = \|\boldsymbol{v}_N\|^2 = 1, \\
& \boldsymbol{v}_i^T\boldsymbol{v}_j \leq w_{ij} + (1 - w_{ij})t.
\end{array} \tag{33}$$

*Proof.* To go from the first problem to the second, we begin by reformulating it as follows:

$$\begin{array}{ll}
\underset{\boldsymbol{u}_1,\ldots,\boldsymbol{u}_N,\boldsymbol{c}\in\mathbb{R}^d}{\text{minimize}} & k, \\
\text{subject to} & \|\boldsymbol{u}_1\|^2 = \|\boldsymbol{u}_2\|^2 = \cdots = \|\boldsymbol{u}_N\|^2 = \|\boldsymbol{c}\|^2 = 1, \\
& \boldsymbol{u}_i^T\boldsymbol{u}_j \leq w_{ij}, \\
& (\boldsymbol{c}^T\boldsymbol{u}_i)^2 \geq \frac{1}{k}.
\end{array} \tag{34}$$

We can also reformulate the second problem as follows, by setting $t = -\frac{1}{k-1}$ and noticing that $t$ is increasing as $k$ increases:

$$\begin{array}{ll}
\underset{\boldsymbol{v}_1,\ldots,\boldsymbol{v}_N\in\mathbb{R}^d}{\text{minimize}} & k, \\
\text{subject to} & \|\boldsymbol{v}_1\|^2 = \|\boldsymbol{v}_2\|^2 = \cdots = \|\boldsymbol{v}_N\|^2 = 1, \\
& \boldsymbol{v}_i^T\boldsymbol{v}_j \leq \frac{w_{ij}k-1}{k-1}, \\
& k > 1
\end{array} \tag{35}$$

Note that we can safely add the constraint $t < 0$ (equivalently $k > 1$ in the second problem without altering the solution. This is because the same argument as in the proof of the unweighted version holds. As $w_{ij} \geq 0$ and $N \leq d$, the solution of the vectors being orthogonal is feasible and will have a $t \leq 0$. This is because one would have $w_{ij} + (1 - w_{ij})t = 0$ for some pair $i, j$ with $w_{ij} < 1$ (otherwise the constraint is trivial), and so $t = -\frac{w_{ij}}{1-w_{ij}} \leq 0$. It is then possible to alter the solution slightly as before to decrease all non-diagonal elements so that $t < 0$ is also satisfied. It is sufficient to show that the problems in equation 34 and equation 35 are equivalent. Following the technique used by Gärtner & Matousek (2012) the regular Lovasz theta problem, we do so by showing the two problems have the same optimal value. Let $p_1$ and $p_2$ be the optimal values of the problems in equation 34 and equation 35, respectively.

To show that $p_1 \geq p_2$, consider an optimal solution $\boldsymbol{u}_1^*, \ldots, \boldsymbol{u}_N^*, \boldsymbol{c}^*$. For the same reasons as in the unweighted case, this solution will give $p_1 \geq 1$, and $p_1 = 1$ only appears if there are no constraints on the inner products. Let us formulate the following matrix $\mathbf{Y}$, with elements:

$$y_{ii} = 1.$$

$$y_{ij} = \frac{1}{p_1 - 1}\left(\frac{\boldsymbol{u}_i^*}{\boldsymbol{c}^{*T}\boldsymbol{u}_i^*} - \boldsymbol{c}^*\right)^T \left(\frac{\boldsymbol{u}_j^*}{\boldsymbol{c}^{*T}\boldsymbol{u}_j^*} - \boldsymbol{c}^*\right)^T = \frac{1}{p_1 - 1}\left(\frac{\boldsymbol{u}_i^{*T}\boldsymbol{u}_j^*}{(\boldsymbol{c}^{*T}\boldsymbol{u}_i^*)(\boldsymbol{c}^{*T}\boldsymbol{u}_j^*)} - 1\right) \quad (36)$$

$$\leq \frac{p_1 w_{ij} - 1}{p_1 - 1}.$$

We can also show that this matrix is PSD; since $y_{ii} \geq y_{ij}, \forall j$, $\mathbf{Y}$ can be written as $\mathbf{Y} = \mathbf{D} + \mathbf{U}^T\mathbf{U} \succeq 0$, where $\mathbf{D}$ is a diagonal matrix with non-negative entries and $\mathbf{U}$ is a matrix with columns equal to $\frac{\boldsymbol{u}_i^*}{\boldsymbol{c}^{*T}\boldsymbol{u}_i^*} - \boldsymbol{c}^*$. Consequently, we can use this $\mathbf{Y}$, to create a feasible solution $\boldsymbol{v}_1, \ldots, \boldsymbol{v}_N$ via Cholesky factorization (i.e. $\mathbf{Y} = VV^T$, whenever $N \leq d$ (the constraints arise from the constraints placed on the matrix $\mathbf{Y}$). This feasible solution has objective value $p_1$ for the second problem. This means that the second problem has optimal value $p_2 \leq p_1$.

To show that $p_2 \geq p_1$, start from an optimal solution of the second problem $\boldsymbol{v}_1^*, \ldots, \boldsymbol{v}_N^*$. Let $\mathbf{Y}^*$ be the matrix with $y_{ij}^* = \boldsymbol{v}_i^{*T}\boldsymbol{v}_j^*$. We note here that this matrix must have at least one eigenvalue equal to 0. If this was not the case, then we would have $\lambda_{min}(\mathbf{Y}^*) > 0$, and we could construct the following matrix:

$$\mathbf{Y}' = \mathbf{Y}^* + \epsilon(\mathbf{I} - \mathbf{1}). \quad (37)$$

where $\mathbf{1}$ is the rank-1 matrix with all of its elements equal to 1. Note that this would strictly decrease all the non-diagonal elements of $\mathbf{Y}^*$, while leaving the diagonal ones intact. Furthermore, $\mathbf{1}$ has $N - 1$ eigenvalues equal to 0, and one of them equal to $N$ (as $\mathbf{1}u = Nu$, where $u$ is the all-ones vector). Thus, since $\mathbf{I} - \mathbf{1}$ is a rank-1 difference from the diagonal, $N - 1$ of its eigenvalues being equal to 1 and one of them being equal to $1 - N < 0$ (given the eigenvalues of $\mathbf{I}$). Thus, we would have, for $\epsilon$ small enough:

$$\lambda_{min}(\mathbf{Y}') \geq \lambda_{min}(\mathbf{Y}^*) + \epsilon(1 - N) \geq 0. \quad (38)$$

Thus, by Cholesky decomposition again, we would have a feasible solution to problem 35, with strictly smaller off-diagonal elements of $y_{ij}$, which is a contradiction. Thus, one of the eigenvalues of $\mathbf{Y}$ must be 0, which means that we can find a vector $\boldsymbol{c}$ which is orthogonal to all $\boldsymbol{v}_1^*, \ldots, \boldsymbol{v}_N^*$. We can then define the vectors:

$$\boldsymbol{u}_i = \frac{1}{\sqrt{p_2}}(\boldsymbol{v}_i^*\sqrt{p_2 - 1} + \boldsymbol{c}). \quad (39)$$

These vectors have:

$$\boldsymbol{u}_i^T\boldsymbol{u}_j = \frac{1}{p_2}((p_2 - 1)\boldsymbol{v}_i^{*T}\boldsymbol{v}_j^* + 1). \quad (40)$$

Thus $\|\boldsymbol{u}_i\|^2 = \frac{1}{p_2}(p_2 - 1 + 1) = 1$ and $\boldsymbol{u}_i^T\boldsymbol{u}_j \leq \frac{1}{p_2}(w_{ij}p_2 - 1 + 1) = w_{ij}$. This means that they form a feasible solution for our problem, with objective value $p_2$. Thus $p_1 \leq p_2$.

Combining all of the above gives us $p_1 = p_2$, making the above problems equivalent.

$\square$

**Theorem A.2** (Theorem 4.2 in the main paper.). *Consider the following formulation of the weighted Lovasz theta problem:*

$$
\begin{aligned}
\underset{\boldsymbol{v}_1,\dots,\boldsymbol{v}_N\in\mathbb{R}^d}{\text{minimize}} \quad & t, \\
\text{subject to} \quad & \|\boldsymbol{v}_1\|^2 = \|\boldsymbol{v}_2\|^2 = \cdots = \|\boldsymbol{v}_N\|^2 = 1, \\
& \boldsymbol{v}_i^T \boldsymbol{v}_j \leq w_{ij} + (1-w_{ij})t,
\end{aligned}
\tag{41}
$$

*as well as the minimization of the second term of the Lovasz theta contrastive loss:*

$$
\begin{aligned}
\underset{\boldsymbol{v}_1,\dots,\boldsymbol{v}_N\in\mathbb{R}^d}{\text{minimize}} \quad & \tau \sum_{i=1}^{N} \log\left( \sum_{j=1}^{N} \exp\left( \frac{\boldsymbol{v}_j^T \boldsymbol{v}_i - w_{ij}}{\tau(1-w_{ij})} \right) \right), \\
\text{subject to} \quad & \|\boldsymbol{v}_1\|^2 = \|\boldsymbol{v}_1\|^2 = \|\boldsymbol{v}_2\|^2 = \cdots = \|\boldsymbol{v}_N\|^2 = 1.
\end{aligned}
\tag{42}
$$

*Minimizing the limit of the objective of the second problem as $\tau \to 0$ is a relaxation of the first problem, if $w_{ij} < 1$.*

*Proof.* Using the property of Log-Sum-Exp to converge uniformly to the maximum of its arguments as $\tau$ goes to $0$, in this limit the objective of the second problem becomes:

$$
\begin{aligned}
\underset{\boldsymbol{v}_1,\dots,\boldsymbol{v}_N\in\mathbb{R}^d}{\text{minimize}} \quad & \sum_{i=1}^{N} \max_{j\neq i} \frac{\boldsymbol{v}_j^T \boldsymbol{v}_i - w_{ij}}{1-w_{ij}}, \\
\text{subject to} \quad & \|\boldsymbol{v}_1\|^2 = \|\boldsymbol{v}_1\|^2 = \|\boldsymbol{v}_2\|^2 = \cdots = \|\boldsymbol{v}_N\|^2 = 1,
\end{aligned}
\tag{43}
$$

(note that we do not argue about the behavior of the minimizing solution of the problem as $\tau \to 0$, but rather just the minimization of the limit of the objective function). We now include auxiliary variables $t_i$, changing the problem into the following:

$$
\begin{aligned}
\underset{\boldsymbol{v}_1,\dots,\boldsymbol{v}_N\in\mathbb{R}^d}{\text{minimize}} \quad & \sum_{i=1}^{N} t_i, \\
\text{subject to} \quad & \|\boldsymbol{v}_1\|^2 = \|\boldsymbol{v}_2\|^2 = \cdots = \|\boldsymbol{v}_N\|^2 = 1, \\
& \boldsymbol{v}_i^T \boldsymbol{v}_j \leq w_{ij} + (1-w_{ij})t_i, \ \forall i \neq j.
\end{aligned}
\tag{44}
$$

This is a relaxation of the following problem:

$$
\begin{aligned}
\underset{\boldsymbol{v}_1,\dots,\boldsymbol{v}_N\in\mathbb{R}^d}{\text{minimize}} \quad & \sum_{i=1}^{N} t_i, \\
\text{subject to} \quad & \|\boldsymbol{v}_1\|^2 = \|\boldsymbol{v}_2\|^2 = \cdots = \|\boldsymbol{v}_N\|^2 = 1, \\
& \boldsymbol{v}_i^T \boldsymbol{v}_j \leq w_{ij} + (1-w_{ij})t_i, \ \forall i \neq j, \\
& t_i = t, \ \forall i,
\end{aligned}
\tag{45}
$$

which is equivalent to precisely the weighted Lovasz theta problem:

$$
\begin{aligned}
\underset{\boldsymbol{v}_1,\dots,\boldsymbol{v}_N\in\mathbb{R}^d}{\text{minimize}} \quad & t, \\
\text{subject to} \quad & \|\boldsymbol{v}_1\|^2 = \|\boldsymbol{v}_2\|^2 = \cdots = \|\boldsymbol{v}_N\|^2 = 1, \\
& \boldsymbol{v}_i^T \boldsymbol{v}_j \leq w_{ij} + (1-w_{ij})t, \ \forall i \neq j.
\end{aligned}
\tag{46}
$$

In this case, the lack of symmetry of the problem does not allow us to immediately argue that all the $t_i$ must be equal. □

## B  TRAINING DETAILS

In our CIFAR experiments, we made use of an A100 GPU to train our models. In the supervised case, our models were trained for 300 epochs, with a batch size of 512, and the same set of hyperparameters as those used in the Supervised Contrastive learning baseline (Khosla et al., 2020). The architecture used in each of the experiments was a ResNet of varying depth, and the projection head used to perform contrastive learning was a two-layer MLP, reducing the dimension of the features to 128. Evaluation was performed by training a linear classifier on top of the model for 10 epochs, and reporting the best accuracy obtained on the test set across all of the epochs. In the unsupervised case, the architecture is the same, but our models where trained using a batch size of 1024, a learning rate and temperature $\tau$ equal to 0.5, and a linear probe trained for 100 epochs over the learned representations (as done in the repository for the code of Khosla et al. (2020))

For our ImageNet-100 experiments, we made use of computing nodes with 4 RTX5000 GPUs. For models trained with cross-entropy loss, we employed a batch size of 256. Moreover, we made use of Momentum Contrast, when training these models. During MoCo training, we maintained a batch size of 256 and a memory bank of size 8192 as in Khosla et al. (2020). We trained each model for 200 epochs using the standard hyperparameter choices found in He et al. (2020).

Our baseline models were trained using the code directly provided by Khosla et al. (2020). The confusion matrix based similarities were obtained based on the predictions of a model trained with Supervised Contrastive Learning.

For the MoCo-v2 baseline, we use the implementation provided by He et al. (2020), with the standard hyperparameters that they use, and learning rate 0.06 and batch size 512, following their linear scaling guidelines. The only alteration we perform are minor necessary modifications for the code to be compatible with CIFAR100.

For the similarity matrices derived by confusion matrices, they were chosen using the confusion matrices derived by the following models:

- For CIFAR, the model used was a ResNet-50 trained via SupCon on CIFAR. The model was trained for 300 epochs, using the same parameters as in Khosla et al. (2020), and achieved similar accuracy to the one reported in that paper.
- For ImageNet-100, the model used was a standard ResNet-50 provided by Pytorch, pretrained on ImageNet.

The overall code is provided as part of the Supplementary material for review purposes, and will be made publicly available upon acceptance. The code is based upon the publicly available code of Supervised Contrastive Learning and Momentum Contrast, and all the relevant licenses are included.

## C  FURTHER EXPERIMENTS

### C.1  QUALITY OF CLIP REPRESENTATIONS

In our experiments above, we derived similarities via inner products using CLIP representations. To show that our method has benefit over simply training a linear model on top of those representations, we train a linear classifier on top of a ResNet-50 CLIP model on CIFAR100, using the preprocessing provided by the CLIP pretrained model. The accuracy we obtained from this evaluation was 67.12%, which is lower than SimCLR and our method trained for 1000 epochs, and even lower than our method trained for 300 epochs, as can be seen in Table 1. As such, we can see that these representations by themselves are not ideally suited for this task, but they can provide benefits to our training procedure, by using our proposed method.

### C.2  CONFUSION MATRIX VISUALIZATION

In this section, we include a visualization of two similarity matrices used in the main paper. We can see the results in Figure 4. It is evident that the confusion matrix based approach is much more selective than the CLIP based one. Combining this with the fact that both the confusion matrix and the superclass similarities outperforms the CLIP based one, as seen in the experimental section, we can infer the result that the more selective the confusion matrix is, the better the results of our method are (which is also intuitively what we expect to happen).

### C.3  EXPERIMENT ON CIFAR10

In Table 5, we can see the results of our method on CIFAR10, using the similarity matrix derived via the confusion matrix of another model. We can see that while our method obtains good accuracy, we cannot get significant improvements over regular supervised contrastive learning. We believe that this is a sensible limitation for our method - due to the very small number of classes, it is highly unlikely that any are that similar to begin with. As such, the main benefit of our method, which is being able to leverage sample similarities via their classes during training does not apply. Nevertheless, since many important tasks contain a much larger number of classes, this is only a small limitation.

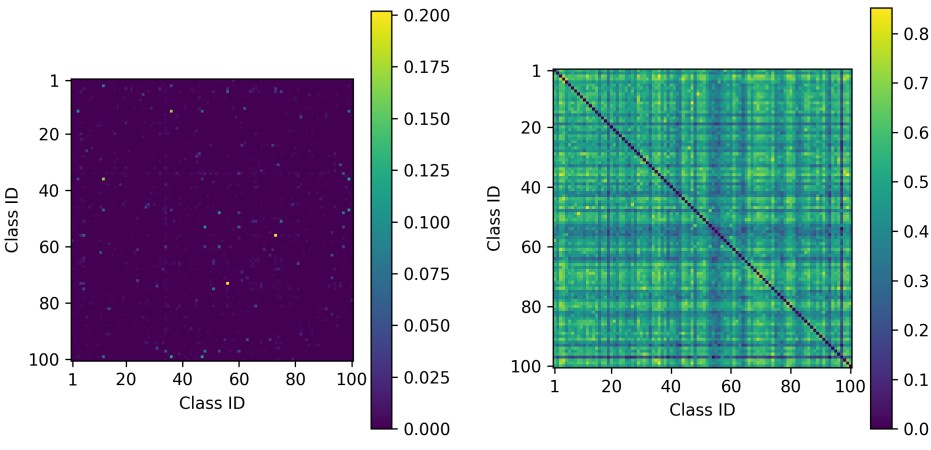

(a) Confusion matrix similarities.                    (b) CLIP based similarities.

Figure 4: **Similarity matrices used in the main paper, supervised setting.** On the left, we see the confusion matrix similarities, and on the right the CLIP based similarities. In both cases, the diagonal has been set to 0 for the purposes of visualization. We can see that while the confusion matrix approach is much more selective in which classes are similar to each other, the CLIP based similarities assign much greater similarities to all classes. This, combined with the results of the relevant experiments, shows the importance of the chosen similarity matrix.

Table 5: **Summary of our results on CIFAR10.** In this case, our method is competitive with SupCon, but there is a limitation due to the fewer number of classes. The numbers with * are taken from Khosla et al. (2020).

|  | CE | SupCon | Ours |
|---|---|---|---|
| ResNet-50 | 95* | **96*** | 95.47 |
| ResNet-34 | 95.15 | **95.28** | 95.07 |
| ResNet-18 | 94.37 | 94.61 | **94.69** |

### C.4 ABLATION ON THE SUPERCLASS SIMILARITIES

As an ablation, we performed the experiments presented in the main paper regarding the superclass similarities on CIFAR, using a different value for the similarity of classes belonging to the same superclass. The results can be seen in Table 6. We can see that we get comparable results with those seen in the main paper, so tuning this hyperparameter may prove useful, in order to improve the performance of the model.

### C.5 ITERATIVE UPDATES TO THE SIMILARITY MATRIX

As an extra experiment, we have included an iterative update scheme to our supervised setting. In this experiment, train a model on CIFAR100 using the superclass similarity matrix $\mathbf{W}$ as a base. Every 50 epochs, we calculate a new similarity matrix $\mathbf{W}'$ via the confusion matrix of the current model, and set $\mathbf{W} = \beta\mathbf{W}+(1-\beta)\mathbf{W}'$, with $\beta = 0.9$ a hyperparameter. After training this model for 300 epochs, we obtained an accuracy of 76.68% on CIFAR100. While this underperforms compared to our methods, it is still competitive and poses an interesting direction for further research.

## D FURTHER DIRECTIONS

Based on all of the above, it is not evident whether there is a single way to obtain the proper similarity matrix, to get the best possible results. As can be seen by our experiments, this design decision

Table 6: **Ablation on superclass similarity.** Our method has comparable results, based on how similar samples from the same superclass but different classes are considered to be (0.5 or 0.8 in this case).

|  | Superclass Similarity 0.5 | Superclass Similarity 0.8 |
| --- | --- | --- |
| ResNet-50 | $77.60 \pm 0.30$ | $77.68 \pm 0.49$ |
| ResNet-34 | $76.55 \pm 0.40$ | $76.35 \pm 0.06$ |
| ResNet-18 | $74.91 \pm 0.15$ | $74.99 \pm 0.32$ |

influences the performance of our models. As such, our work opens interesting research directions in identifying good similarity metrics between samples in contrastive learning.

