# OpenReview forum: "Lovasz Theta Contrastive Learning"
_ICLR.cc/2023/Conference — Submitted to ICLR 2023_

### Official Review · Reviewer_fWo7 · 2022-10-24

**Confidence:** 4
**Correctness:** 3
**Technical Novelty And Significance:** 3
**Empirical Novelty And Significance:** 2
**Recommendation:** 5

**Clarity, Quality, Novelty And Reproducibility:**

Clarity: see above.
Quality: the paper is reasonably well written with supporting experiments
Novelty: the connection to Lovasz theta problem is novel
Reproducibility: the authors attach code for reproducibility check

**Strength And Weaknesses:**

Strengths:
1) The connection between the negative term of InfoNCE and Lovasz theta problem from graph theory is novel. The generalized version also makes sense, since the weight $w_{ij}$ represents the similarity between samples $i$ and $j$.
2) The authors test the Lovasz generalization to both supervised and unsupervised cases, and show that the linear fine-tuning accuracy can be improved over SimCLR and SupCon.
3) The supplementary code is attached for reproducibility.
4) The similarity function can be defined from existing feature encoders such as CLIP. This is a highlight of the current work.

Weaknesses:
There are indeed some places where this draft can be improved.
1) Minor issues of writing/typos: some informal abbreviations: "doesn't", "s.t."(eq. (3)), "don't", "aren't"; parenthesis in Theorem 3.1
2) The connection part could be written more clearly. Some insights from the appendix can be summarized in the main text. For example, the Jensen's inequality of the negative term leads to a constrained optimization problem w.r.t. the inner products, and thus the Lovasz theta problem.
3) The 2nd assumption in Sec 4.1 doesn't seem necessary. One can just say that the negative term relates to the Lovasz theta problem.
4) The comparison between SimCLR and the proposed method is a bit unfair: the Lovasz theta method uses CLIP as some prior knowledge, but this is not used in SimCLR. Also, only SimCLR and SupCon are compared while there are many competitive contrastive learning methods such as MoCo (v1, v2, v3) and SimCLRv2.
5) The theoretical analysis is not strong enough. Although the paper finds equivalence and extension of InfoNCE, it doesn't show any theory of convergence or generalization property.


**Summary Of The Paper:**

This work points out an interesting connection between Lovasz theta function and contrastive learning. More concretely, the negative term in the InfoNCE loss corresponds to the Lovasz theta problem, with zero weights. From this connection, the authors are able to extend the InfoNCE loss to a broader class and the experiments show some improvement over SimCLR.

--post rebuttal--

Thanks to the authors for the rebuttal. After the revision, the paper looks better. However, the main contribution seems less novel to me. The connection Theorem 4.1 is somehow known in contrastive learning, for example, Theorem 5 (revised version) in https://sslneurips21.github.io/files/CameraReady/f-MICL-NeurIPS-workshop.pdf (the exact form is slightly different but it can extend to the current paper). I think the main novelty is the extension to different similarity weights, but the comparison with baseline algorithms is limited (only MoCo, SimCLR and SupCon). More SSL baselines are needed, e.g., SimSiam, BYOL, SwAV, etc. Also, the similarity term $w_{ij}$ needs to be discussed more.

For the fair comparison part, I don't understand the author's comment. Why can't we use CLIP as a pre-trained model for SimCLR? In this way, both Lovasz and SimCLR would have the same prior knowledge.

In summary, despite being interesting, I think this paper does not have enough contribution to be accepted, and I disagree with Reviewer 4DeW that it should be given the highest score of ICLR.

**Summary Of The Review:**

In summary, my recommendation of the paper is mixed: on the one side, the connection with Lovasz theta problem is defintely interesting and opens up a new approach for finding similarity function; on the other side, this connection is not well explained in the main paper and more empirical comparison is needed.

---

> ### Author Response · Authors · 2022-11-18
> **Response to Reviewer fWo7**
>
> We thank the Reviewer for appreciating the novelty of the connection to the Lovasz theta problem, and we are also grateful for the Reviewer pointing out how we can extend contrastive learning using similarities derived by models such as CLIP. Below are responses to the Reviewer's comments.
>
> **Minor issues of writing/typos: some informal abbreviations: "doesn't", "s.t."(eq. (3)), "don't", "aren't"; parenthesis in Theorem 3.1**
>
> We thank the Reviewer for pointing out these issues. We have fixed them in the revised version.
>
> **The connection part could be written more clearly. ... the Lovasz theta problem.**
>
> We thank the Reviewer for their comment. We have included some intuition regarding the proof of Theorem 4.1 which shows the connection between the two problems in Section 4.1.
>
> **The 2nd assumption in Sec 4.1 doesn't seem necessary. One can just say that the negative term relates to the Lovasz theta problem.**
>
> We thank the Reviewer for their suggestion. Our reasoning behind formulating this approach as an assumption is to leave open the possibility of generalizing our statement by simply lifting the assumption in future work, as opposed to having to rephrase the entire statement. We have clarified that we consider the negative term in Theorem 4.1 while also keeping the assumption for this reason, but we are also happy to remove this assumption, if this conveys the message of our work more cleanly to the reader.
>
> **The comparison between SimCLR and the proposed method is a bit unfair: the Lovasz theta method uses CLIP as some prior knowledge, but this is not used in SimCLR. Also, only SimCLR and SupCon are compared while there are many competitive contrastive learning methods such as MoCo (v1, v2, v3) and SimCLRv2.**
>
> We thank the Reviewer for their comments. We want to point out that the ability to use prior knowledge in our method is precisely one of our main contributions. This makes it an extension over other methods, and allows it to be applied in the setting where we have insight on class/sample similarities for the given task.
>
> Following the Reviewer’s suggestion, we examined an extra baseline (MoCo-v2, with the implementation provided from the official MoCo github repository). We used the hyperparameters provided by that repository, with only a change in batch size to 512 and learning rate 0.06 per their linear scaling guidelines. With that, we were able to get up to 64.43% accuracy in ResNet-50 (see Table 1 for details). We also want to note that our proposed method directly alters the loss function used, which is somewhat orthogonal to the precise training method and augmentations used (such as MoCo or SimCLR).
>
> **The theoretical analysis is not strong enough. Although the paper finds equivalence and extension of InfoNCE, it doesn't show any theory of convergence or generalization property.**
>
> We thank the Reviewer for their comment. While our work does not include theoretical results on convergence and generalization, and these are very interesting research directions to take, we believe that this is outside the scope of the current paper. We comment here that our loss function is convex with respect to the inner products of the representations (as long as $N \leq d$) so regarding convergence, standard results using optimization methods such as SGD should still hold with respect to the outputs (but possible not with respect to the model parameters, over which the problem is non-convex if the model is complex).

---

### Official Review · Reviewer_4DeW · 2022-10-25

**Confidence:** 4
**Correctness:** 4
**Technical Novelty And Significance:** 4
**Empirical Novelty And Significance:** 3
**Recommendation:** 6

**Clarity, Quality, Novelty And Reproducibility:**

The work is presented clearly with necessary introduction of current work and explanation on how it builds upon the same. The contribution is novel and original as it bridges a previously unexplored bridge between contrastive learning and graph theoretical approaches and semi-definite programming. The paper provides all analytical proofs stated. It also provides code and selection of hyperparameters to help with reproducibility. However, the paper does not refer to any previous runs / logbooks of trails. This will provide a way for readers to understand the way the problem was approached.

**Strength And Weaknesses:**

The paper introduces a new loss function and provides proof that the minimization of certain terms in the proposed loss function is a relaxation of the weighted Lovasz theta function. To do this, they start with a very widely-used contrastive loss function - InfoNCE. They prove that minimizing InfoNCE is the same as minimizing a certain formulation of the Lovasz theta function. This provides a strong basis for the theory of contrastive learning. Additionally it provides a way to include similarity metrics into contrastive methods.

However, the results showed minor improvements in performance from SimCLR. It is also prudent to compare the method with other SoTA self-supervised learning methods that learn fine-grained granular features like MUlti-Granular Self-supervised learning (Mugs). We advise the authors to do this as it will help measure the effect of using similarity metrics in this particular loss formulation on performance.


**Summary Of The Paper:**

The paper establishes a connection between the Lovasz theta function and contrastive learning. Specifically, with InfoNCE - a type of contrastive loss function used for self-supervised learning. They define a novel loss function - Lovasz theta contrastive loss based on a weighted graph representation of the similarities between examples..
The loss function is evaluated with various ways of quantifying sample similarity. The method outperforms existing model SimCLR in both supervised and unsupervised contrastive learning. Previous works introduce the notion of using graph representation with edge weights equal to the probability that two samples correspond to views of the same sample. Current work uses that representation to perform contrastive learning. Under certain choices for model dimensionality and batch size, the paper shows an equivalence between minimizing InfoNCE loss and the Delsarte formulation of the Lovasz theta problem.
This equivalence is shown in both supervised and unsupervised cases. The equivalence is used to incorporate degree-of-similarity information into contrastive learning. So this method is a generalization of regular contrastive learning. In unsupervised cases, the method outperforms regular SimCLR pretraining on CIFAR100. In the supervised case, the method outperforms simple cross entropy loss and supervised contrastive learning on CIFAR100 and ImageNet-100. The method can be used with any similarity measure to create a problem specific contrastive loss.
As a proof of improvement, the work plots the effect of blending regular contrastive learning with the degree-of-similarity aware method they created. A parameter λ = 0 (regular SupCon) to λ = 1 (their method) is used to blend the effect of setting the class similarity matrix as C’ = λC + (1 − λ)I. It shows an overall improvement in performance as λ increases.
The work links the field of contrastive learning to classic graph theory and semi-definite programming. By doing this, it tries to provide a direction to understand the underpinnings of contrastive learning.


**Summary Of The Review:**

The paper provides an contribution of bridging the field of contrastive learning with Lovasz theta functions and graph theory. We think it will pave the way for incorporating more similarity metrics into contrastive learning methods. The paper provides excellent explanations of contrastive learning functions therefore it is valuable for a wide audience.

---

> ### Author Response · Authors · 2022-11-18
> **Response to Reviewer 4DeW**
>
> We are very grateful to the Reviewer for appreciating our work, its originality and how it builds upon contrastive learning as a whole. We are also thankful for their comment on how our work builds a strong basis for the theory of contrastive learning. Below are responses to some of the Reviewer's comments.
>
> **It is also prudent to compare the method with other SoTA self-supervised learning methods that learn fine-grained granular features like MUlti-Granular Self-supervised learning (Mugs). We advise the authors to do this as it will help measure the effect of using similarity metrics in this particular loss formulation on performance.**
>
> We thank the Reviewer for their suggestion. MUGS [C] is indeed related to the concept of learning good representations for images, using features at different granularities. As it is built using features from ViT type models, we were unfortunately unable to directly compare against it due to limited computational resources available for running these additional experiments within the limited timeframe of the response period, but we included it as part of our related work.
>
> **However, the paper does not refer to any previous runs / logbooks of trails. This will provide a way for readers to understand the way the problem was approached.**
>
> We thank the Reviewer for their comment. The way this problem was approached was by first examining the relationship of the InfoNCE loss and the Lovasz theta problem, using the proofs provided in our work. We then extended this connection by introducting our novel form of contrastive loss, which we then evaluated experimentally.
>
> **References**
>
> [C]: Pan Zhou, Yichen Zhou, Chenyang Si, Weihao Yu, Teck Khim Ng, and Shuicheng Yan. Mugs: A multi-granular self-supervised learning framework. arXiv preprint arXiv:2203.14415, 2022.

---

### Official Review · Reviewer_NuU3 · 2022-10-25

**Confidence:** 4
**Correctness:** 3
**Technical Novelty And Significance:** 4
**Empirical Novelty And Significance:** 3
**Recommendation:** 6

**Clarity, Quality, Novelty And Reproducibility:**

Clarity:
1. Based on the equivalence to Lovasz theta, I wonder if more can be said about the learned representations, e.g., the geometry of learned representations and even their effect on generalization in downstream task.
2. Clarification on the quality of CLIP representations and confusion matrix is needed.

Novelty:
1. The main insights of the connection between contrastive learning and Lovasz theta, and leveraging the connection to derive a generalized version of loss are novel and can be important to the field. The idea of incoporating prior knowledge of similarity into contrastive learning is also novel.

Seems the proposed loss function explicitly encourages non-uniform pair-wise similarity. It would also be interesting to see if this can help address some other problems in supervised contrastive learning, e.g., class-collapse.

**Strength And Weaknesses:**

Strength:
1. The paper provides a new view of contrastive learning by showing its connection to the Lovasz theta problem, which may help further the understanding of contrastive learning.
2. The proposed loss can be used to boost contrastive learning with any prior knowledge on similarity between examples.
3. The proposed method can be seen as a new way of using information from pretrained models and transfering the useful knowledge in them to new relevant tasks in the context of contrastive learning. It is more flexible than finetuning since it does not require the model architecture to be the same as the pretrained one.

Weakness:
1. Theorem 4.1 only shows that the negative pair part in the loss is equivalent to Lovasz theta. The positive pair part is nullified by the assumption of the single positive case. This makes the result less informative considering that the positive part can play an important role. For example, [1] has shown that it encourages alignment between positive pairs and has certain interaction with the negative pair part which corresponds to uniformity.
2. It is unclear what the quality of representations given by CLIP is. I think most likely it would not be very high, but it is still important to clarify since otherwise there is always a possibility that the CLIP representations are already good enough and it is trivial to beat the baseline by aligning the similarity output by the model with the good-enough representations'.
3. More clarification is needed on the confusion matrix used in experiments. The authors said it is given by a model pretrained using supcon. But what is the setup for the pretraining and how long has the model been trained? what's the linear evaluation accuracy on its representations? Without knowing the answers to the above questions, it is hard to judge if the corresponding experimental results are meaningful.

[1] Wang, Tongzhou, and Phillip Isola. "Understanding contrastive representation learning through alignment and uniformity on the hypersphere." International Conference on Machine Learning. PMLR, 2020.

**Summary Of The Paper:**

The authors established a connection between contrastive learning and the Lovasz theta problem. They show that with certain assumption the original loss function corresponds to Lovasz theta with an empty similarity graph. Then, by considering Lovasz theta on weighted graphs, they designed a new loss function that incoporates prior knowledge of similarity between examples. Experiements are performed to show that their method can be used to improve both unsupervised and supervised contrastive learning.

**Summary Of The Review:**

The contribution of the paper is two fold (1) it showed the connection between contrastive learning and Lovasz theta, which is a novel viewpoint of contrastive learning (2) and subsequently proposed a loss function that incoporates prior knowledge of similarity. However, the connection to Lovasz theta doesn't yet provide much insights into properties of the learned representations, since there is no further discussion about this in the paper. This may limit its theoretical contribution. Some details are missing in the experiment section. Also, I wonder if one should also expect that adaptively adding similarity information obtaind from the representations given by the current model every certain epochs during training can also improve the performance (i.e., training -> updating similarity matrix using current representations -> training -> updating -> training ...)

---

> ### Author Response · Authors · 2022-11-18
> **Response to Reviewer NuU3 (Part 1)**
>
> We thank the Reviewer for their comments, and we are grateful for them pointing out how our work provides a novel perspective on contrastive learning. We are also thankful for their comment on how our technique can be used to transfer knowledge from pretrained models to new ones. Below are detailed responses to the Reviewer's comments.
>
> **Theorem 4.1 only shows that the negative pair part in the loss is equivalent to Lovasz theta. ... the negative pair part which corresponds to uniformity.**
>
> We thank the Reviewer for their comment. Indeed, [B] has shown that the positive term encourages alignment between representations, while the negative term encourages uniformity. We believe that further extending this negative term and demonstrating its links to the Lovasz theta is a significant theoretical contribution. Of course, incorporating the positive part in this formulation is an important direction, which we will consider for extensions on future work. We invite the Reviewer to see our updates at the end of Section 4.1 on discussion on this topic.
>
> **It is unclear what the quality of representations given by CLIP is. ... the model with the good-enough representations.**
>
> We thank the Reviewer for their insightful suggestion. Following this, we have trained a linear classifier on top of the ResNet-50 CLIP model, using its provided preprocessing values. The accuracy we received from this was 67.12%, which is lower than SimCLR and our method trained for 1000 epochs, as well as lower than our method trained for 300 epochs. As such, we can see that these representations by themselves are not ideally suited for this task. We refer the Reviewer to Appendix C.1 in the revised paper for discussion on this.
>
> **More clarification is needed on the confusion matrix used in experiments. ... if the corresponding experimental results are meaningful.**
>
> Regarding the similarity obtained by the confusion matrix for CIFAR100, this is derived by the baseline SupCon model for ResNet-50, trained using the code provided by the authors of SupCon for 300 epochs, with the rest of hyperparameters being the same. Its linear evaluation accuracy is close to the one reported by the authors of that paper. For ImageNet-100, these similarities are derived by a pretrained ResNet-50 provided by Pytorch. We have clarified this in the revised paper, in Section 5 and Appendix B. Furthermore, we want to highlight that the confusion matrix is only one of the many ways one could try to obtain the similarities. In Table 2, we experiment with different choices – in the case of CIFAR100 the best performing choice is using the superclasses provided by the dataset itself.
>
> **Based on the equivalence to Lovasz theta, I wonder if more can be said about the learned representations, e.g., the geometry of learned representations and even their effect on generalization in downstream task.**
>
> We thank the Reviewer for their suggestion. The equivalence with Lovasz theta corroborates the uniformity effect of the negative term, as shown by [B]. Both investigating the geometry of the learned representations and the generalization capabilities of our method are interesting directions for future work.
>
> **Clarification on the quality of CLIP representations and confusion matrix is needed.**
>
> We refer the Reviewer to Section 5, Appendix B and Appendix C.1 for discussion on these topics.
>
> **Seems the proposed loss function explicitly encourages non-uniform pair-wise similarity. It would also be interesting to see if this can help address some other problems in supervised contrastive learning, e.g., class-collapse.**
>
> We thank the Reviewer for their interesting suggestion. Indeed, our formulation explicitly allows for similarities to differ between samples and classes. This asymmetry may be helpful to alleviate collapse issues when training contrastive models. While further work needs to be done to examine this, we are grateful for the interesting research direction.

---

> ### Author Response · Authors · 2022-11-18
> **Response to Reviewer NuU3 (Part 2)**
>
> **I wonder if one should also expect that adaptively adding similarity information ... can also improve the performance**
>
> Regarding the iterative experiment, we thank the reviewer for their insightful suggestion. In this direction, we have constructed an experimental setup where we train our model for 300 epochs and iteratively update the similarity matrix each time, based on the confusion matrix derived from the model every 50 epochs. This iterative update is done via a convex combination of the previous similarity matrix and the one derived from the confusion matrix of the current model (this is done for the supervised case). The accuracy we got after 300 epochs of training is 76.68%. While this is competitive with SupCon, it is not as good as the rest of our methods. However, it may be possible to get better results with further tuning of the update schedule. We have included this extra experiment in Appendix C.5.
>
> **References**
>
> [B]: Tongzhou Wang and Phillip Isola. Understanding contrastive representation learning through alignment and uniformity on the hypersphere. In International conference on machine learning. PMLR, 2020

---

### Official Review · Reviewer_HNxb · 2022-11-01

**Confidence:** 4
**Correctness:** 2
**Technical Novelty And Significance:** 2
**Empirical Novelty And Significance:** 2
**Recommendation:** 3

**Clarity, Quality, Novelty And Reproducibility:**

- Clarity: This paper is well-written and easy to follow.
- Quality: The proof may not be rigorous. The formal version of the main theorem should not be hidden in the appendix.
- Novelty: The novelty is limited since it only considers the negative sample part of InfoNCE. However, the alignment of positive samples may be more important for contrastive learning, since several methods such as byol/simsiam only consider the alignment and can achieve good performance. At least, the author should discuss the importance of the negative sample part.

**Strength And Weaknesses:**

Strength:
- The view of Lovasz theta problem is interesting.
- The proposed loss achieves significant improvement on empirical performance.

Weaknesses:
- This paper can only deal with the negative-sample part of InfoNCE and ignores the positive-pair part. The quality of a representation should be affected by these two factors together and these two factors are not independent.

- The relaxation (7) of the original problem (2) is not intuitive. Why do the authors explain the upper bound of  $u_i^T u_j$ as a similarity? Is the minimum similarity equal to zero? In the unsupervised case where CLIP is applied, the similarities could be [-1,1]? In summary, the authors should explain more about how to interpret the weight $w_{ij}$ in the weighted Lovasz theta problem as a similarity.

- I carefully read the proof in the appendix. I find that (12) should include another constraint that $k>1$. Otherwise, (11) and (12) are not equivalent. Also, (15) should include the constraint that $t<0$. For example, (15) always has a feasible solution of $t=1$ while (11) may not have any feasible solution (e.g., $E$ is empty and $d<N$). Similarly, (31) also needs $t<0$. But this is weird: Since $w_{ij}>0$, the upper bound of $v_i^Tv_j$ becomes the weighted average of $1$ and a negative value $t$.


**Summary Of The Paper:**

This paper studies the InfoNCE loss widely used in contrastive learning from the view of Lovasz theta function. In particular, regardless the positive-pair term of InfoNCE, minimizing InfoNCE corresponds to solving the Lovasz theta function. Inspired by this, the authors relax the constraint of Lovasz theta function beyond orthonormality, and revise the InfoNCE accordingly. Experiments show that the proposed weighted InfoNCE achieves an improvement of 1% in the supervised case and up to 4% in the unsupervised case.

**Summary Of The Review:**

Overall, the idea is interesting and novel. The empirical performance is impressive. However, this is an unready paper. A number of issues need to be closed, especially the rigorousness of the theory part and how to interpret the weight $w_{ij}$ as a similarity. This paper needs to be further polished before acceptance.

---

> ### Author Response · Authors · 2022-11-18
> **Response to Reviewer HNxb (Part 1)**
>
> We are grateful to the Reviewer for finding our work interesting and acknowledging its empirical benefits, as well as finding our work well-written and easy to follow. Below are responses to the Reviewer's concerns.
>
> **This paper can only deal with the negative-sample part of InfoNCE and ignores the positive-pair part. The quality of a representation should be affected by these two factors together and these two factors are not independent.**
>
> We thank the Reviewer for their comment. It is true that in the case of multiple positive pairs, the minimizer of the loss is affected. The assumption of a single positive pair is only used to derive the theoretical connection between the relaxed Lovasz Theta problem and the second term of our newly introduced loss. This connection leads to improved experimental performance, as we show in our Experimental Section. In practice, we also use the first term of our loss (that corresponds to multiple positive pairs), as done in prior works.
>
> **The relaxation (7) of the original problem (2) is not intuitive ... Lovasz theta problem as a similarity.**
>
> We thank the Reviewer for their comment. The upper bound can be thought of as being influenced by the similarity of two images. This is due to the Lovasz formulation on the underlying weighted graph – the weights of the edges of this graph can be interpreted as similarities, which show how strongly the two samples are connected. The interpretation of these weights as similarities is a natural extension of the Lovasz problem on regular graphs – if the weight approaches 0, then these samples are not connected and the corresponding constraints exist in the Lovasz theta problem. On the other hand, if the weight approaches 1, then these samples are perfectly connected, and the corresponding constraint on their inner product $v_i^T v_j \leq w_{ij} = 1$ becomes trivial. Any value $w_{ij} \in (0,1)$ is essentially a varying degree of similarity – the less similar the sample are, the lower the upper bound becomes, and the more spread apart the representations need to be, in order to satisfy it. This is a natural interpretation of similarity between the samples corresponding to the vertices of the graph, with less similar samples requiring representations that are further apart. Having the minimum similarity be equal to $0$ makes the problem consistent with the Lovasz theta formulation. We have included discussion on this right before Lemma 4.1 in the revision.
>
> We thank the Reviewer for pointing out that in the unsupervised case, the CLIP "similarities" might take negative values. It is true that we interpret the weights as similarities of the samples, as explained above. Interestingly, this is not far from what we observe in practice, even for the CLIP experiments: we noticed that the inner products between the elements of CIFAR100 were always positive, implying that CLIP gathers the embeddings in a small cone. Regardless, we explicitly added a cutoff on the similarities derived by CLIP (forcing them to be positive) and we didn’t observe a significant difference in our results (which is consistent with our observation). We included this in Table 1 in the paper.

---

> ### Author Response · Authors · 2022-11-18
> **Response to Reviewer HNxb (Part 2)**
>
> **I carefully read the proof in the appendix ... and a negative value $t$.**
>
> We really appreciate that the Reviewer carefully read our proof. Under the assumption that $N \leq d$ (the setting under which we operate further in the paper, and under which the [A] define the Lovasz theta problem in the first place), the conditions that $k > 1$ and $t < 0$ are implied at the optimal points of their respective problems. In particular:
>
> - In (11) ((13) in the revised version), with $N \leq d$, the problem always has a feasible solution (all $u_i$ being orthogonal) and we have $1 \leq \frac{1}{(c^Tu_i)^2} \leq k$ as $0 \leq (c^T u_i)^2 \leq 1$, due to the vectors being unit norm. Note that $k = 1$ can only happen if all vectors are aligned, which means that there must be no constraint $u_i^T u_j \leq 0$. This means that the second problem will have no constraint $v_i^T v_j \leq t$, thus $t \to -\infty$, which is consistent with $t = -\frac{1}{k-1}$.
>
> - In (15) ((17) in the revised version), the problem again has a feasible solution where all $v_i$ are orthogonal, and that solution has $t = 0$. From this solution, another feasible one can be derived with $ t < 0$. This means that, at the optimal point, $t < 0$.
>
> In (12) ((14) in the revised version), for completeness the condition $k > 1$ is included, but this is only important for the proof to go from (11) to (15) (as can be seen by the statement of the theorem in the main paper). As such, we have clarified the necessity of $N \leq d$ further before in the paper, and have clarified the steps of the proof according to the above. We once again thank the Reviewer for pointing this out and for the constructive feedback. We invite the Reviewer to read the revised paper.
>
> For similar reasons, (31) ((33) in the revised version) does not require $t < 0$ (since a feasible solution can always be found with t = 0) – only the intermediate problem requires this additional constraint without affecting the rest of the proof or the results.
>
> Furthermore, we want to point out that the fact that in (31) the upper bound is a convex combination of $1$ and a negative value $t$ is not contradictory with the rest of our theory – at the optimal point, representations of different vectors may have any sign as long as the bound is respected. The bound indeed interpolates between $1$ (perfect similarity, so the constraint is trivial) and $t$ (zero similarity, so the constraint is the same as regular Lovasz theta). The effect is that the lower the $w_{ij}$, the less similar the samples are and the more difficult the constraint is to satisfy. We are happy to explain further, if the Reviewer finds that this bound contradicts another part of our theory.
>
> **The proof may not be rigorous. The formal version of the main theorem should not be hidden in the appendix.**
>
> We thank the Reviewer for their comment and invite them to view the proof in the revised paper. Per their suggestion, we have clarified parts of the proof and have moved the formal versions of Theorem 4.1 and Theorem 4.2 to the main paper.
>
> **Novelty: The novelty is limited since it only considers the negative sample part of InfoNCE. ... the authors should discuss the importance of the negative sample part.**
>
> We thank the Reviewer for their comments. We agree that there are several methods that examine only the positive part and obtain good results. At the same time, in the simple formulation of the InfoNCE loss, the negative samples play an important role, encouraging uniformity across the sample representations. Our work provides an alternative use of the negative samples, by incorporating similarity information for the problem. This is an interesting direction that allows us to incorporate domain information directly, which is not present in other methods. We invite the Reviewer to see the end of Section 4.1 for further discussion on this topic.
>
> **References**
>
> [A]: Bernd Gartner and Jiri Matousek. Approximation algorithms and semidefinite programming. Springer Science & Business Media, 2012.

---

> ### Comment · Reviewer_HNxb · 2022-12-09
> **Thanks for the response**
>
> Thanks for the response!
>
> 1. The positive-sample part and negative-sample part of InfoNCE affect the quality of the encoder together. This paper tries to reformulate the negative-sample part as Lovasz theta problem and revise the negative-sample part of InfoNCE. I am still concerned about the rationality of doing so. That's a bit like saying that factor A and factor B together lead to a good result. One somehow revises factor B from some new perspective (only a relaxation of factor B but not directly aiming at the final goal). Then can we say that factor A and new factor B together should lead to a better result? Besides, I think only revising the negative-sample part of InfoNCE is a minor improvement. Instead, if this paper could also include the positive-sample part in the Lovasz theta framework and provide a new understanding of how these two parts interact, it would be much better.
>
> 2. The proposed method requires so-called similarity between each pair of samples, and this paper uses CLIP in an **unsupervised case**. This is very unfair to compare with InfoNCE, since CLIP itself already performs better than the self-supervised methods using InfoNCE.
>
> Based on the above,  I would like to keep my score. Thanks for the clarification.

---

### Author Response · Authors · 2022-11-18
**General Comment**

We thank the Reviewers for their interest in our work and for their insightful comments. We have included detailed responses to each Reviewer's comments below the respective reviews, and have also provided a revised version of our paper.

---

> ### Comment · Area_Chair_aJKH · 2022-12-07
> **Question about experiments**
>
> Dear authors,
> Thanks for your efforts in answering the reviewers' comments.
>
> (1) A new question raised during the discussion with the reviewers, about using the CLIP representations. In particular, the CLIP paper (https://arxiv.org/pdf/2103.00020.pdf) claims that their ResNet50 representations of CIFAR100 evaluated using linear probe are much better (70.3%) than what you reported in the rebuttal (67.12%). The concern is that whether using superior CLIP representations in Eq. (11) biases the learned representations, and in the best case provides as good representations as that of CLIP (i.e. 70.3% in Table 1). I appreciate your clarification on this.
>
> (2) The other question is whether the main boost in the performance comes from weighting the negative pairs in the CL loss (e.g. using CLIP similarities to weight negative pairs in SimCLR loss) or if Eq (11) provides any other advantage? Empirical confirmation of this would be very helpful.
>
> (3) The reviewers would also like to see if (2) can boos the performance of other CL methods. However, considering the limited amount of time this is not the highest priority.
>
> Thank you, AC

---

> > ### Author Response · Authors · 2022-12-08
> > **Re: Question about experiments**
> >
> > Dear AC,
> >
> > Thank you very much for your comments. We include our responses below.
> >
> > 1) We obtained a score of 67.12% with the CLIP representations by using the same implementation (linear model provided by pytorch) and hyperparameters we used in all our experiments. In contrast, as explained in page 38, section A.3 in the CLIP paper, the authors used an L-BFGS solver (as provided by scikit-learn) with a hyperparameter sweep for L2 regularization to get the 70.3% score. Originally, we did not explore separate tuning for the linear classifier in our method, but with some exploration on this hyperparameter, the learning rate and removing the augmentations for the training of the linear classifier, we were able to obtain up to 70.92%. It is possible that further tuning on the rest of the hyperparameters (for example, learning rate scheduling) may improve this further.
> >
> > 2) To clarify, while our Lovasz Contrastive loss (eq. 11) changes the influence of each of the exponential terms of the negative weights, it is not a multiplicative reweighting, but a more complicated rescaling, since the weights $w_{ij}$ are also in the denominator. We experimented with removing $w_{ij}$ from the denominator in the unsupervised setting, and doing so resulted in a drop in accuracy to 63.19% (from 70.92% with tuning). This shows that the formulation via the Lovasz Theta provides useful information to the model.
> >
> > 3) Regarding extension to other methods, this is an interesting topic and we expect that other contrastive learning methods can also benefit from additional information represented by soft similarities. However, this needs to be further explored.

---

### Decision · Program_Chairs · 2023-01-20

**Decision:**

Reject

**Justification For Why Not Higher Score:**

Explained in part 1.

**Justification For Why Not Lower Score:**

N/A

**Metareview: Summary, Strengths And Weaknesses:**

The paper shows that minimizing the InfoNCE loss widely used for contrastive learning corresponds to solving the Lovasz theta function. Empirically, the authors show that the proposed weighted InfoNCE achieves an improvement of 1% in the supervised case and up to 4% in the unsupervised case.

A meeting was held to discuss the paper in detail. The reviewers and I agree that the theoretical formulation is indeed very interesting and novel.

However, the first concern is that the formulation does not capture the effect of the positive pairs which is shown to be crucial for the performance of CL. Besides, empirically, the paper shows that the proposed method outperforms SimCLR and SupCon. But, as I mentioned in my comment to the authors, from the current experiments it is not clear if the performance boost is due to the help of the supervision from CLIP or is due to the contrastive loss proposed by the authors. For a fair comparison, the authors may compare performance of different contrastive methods while incorporating the same similarity information from CLIP in a naive way (e.g. by weighting the negative pairs by some function of their similarity), with their proposed formulation. This will confirm the effectiveness of the proposed formulation and will significantly strengthen the paper.

Therefore, while the paper has a great potential, it may not be ready to be published in its current format.